# Mapping cryptic phosphorylation sites in the human proteome

Dino Gasparotto[1,2,12], Annarita Zanon[1,3,4,12], Valerio Bonaldo[1,12], Elisa Marchiori[1], Massimo Casagranda[1], Erika Di Domenico[1], Laura Copat[1,2], Tommaso Fortunato Asquini[1], Marta Rigoli[1], Sirio Vittorio Feltrin [1], Nuria Lopez Lorenzo[5], Graziano Lolli [1], Maria Pennuto [6,7], Jesùs R Requena[5], Omar Rota Stabelli[1,8,9], Giovanni Minervini[3], Cristian Micheletti [10], Giovanni Spagnolli[1,2], Pietro Faccioli [4,11✉] & Emiliano Biasini [1✉]

## Abstract

Advances in computational and experimental methods have revealed the existence of transient, non-native protein folding intermediates that could play roles in disparate biological processes, from regulation of protein expression to disease-relevant misfolding mechanisms. Here, we tested the possibility that specific post-translational modifications may involve residues exposed during the folding process by assessing the solvent accessibility of 87,138 post-translationally modified amino acids in the human proteome. Unexpectedly, we found that one-third of phosphorylated proteins present at least one phosphosite completely buried within the protein's inner core. Computational and experimental analyses suggest that these cryptic phosphosites may become exposed during the folding process, where their modification could destabilize native structures and trigger protein degradation. Phylogenetic investigation also reveals that cryptic phosphosites are more conserved than surface-exposed phosphorylated residues. Finally, cross-referencing with cancer mutation databases suggests that phosphomimetic mutations in cryptic phosphosites can increase tumor fitness by inactivating specific onco-suppressors. These findings define a novel role for co-translational phosphorylation in shaping protein folding and expression, laying the groundwork for exploring the implications of cryptic phosphorylation in health and disease.

Keywords Protein Folding; Post-translation Modification; Protein Phosphorylation; Co-translational Phosphorylation; Cryptic Phosphosites
Subject Categories Post-translational Modifications & Proteolysis; Proteomics; Translation & Protein Quality

## Introduction

Proteins are canonically regarded as physiologically functional in their native conformation. However, the actual landscape of protein conformations is populated by various dynamically interconverting metastable species whose abundance and persistence are determined by the interplay of equilibrium thermodynamic processes, such as the thermally activated transitions between metastable conformations, and non-equilibrium phenomena, for example, the polypeptide synthesis or degradation (Frauenfelder, 2002). The fundamental process where non-native conformers hold unquestioned relevance is protein folding, characterized by the conversion of a newly synthesized, unstructured polypeptide chain to its native three-dimensional conformation. The physiochemical mechanism underlying this process has long been debated, starting from Anfisen's seminal work (Leventhal, 1969; Anfinsen, 1973; Onuchic 1997; Ferreiro, 2018). Theoretical simulations and in vitro biophysics experiments consistently support the hypothesis that the fast refolding mechanism of isolated small globular proteins (i.e., consisting of a few tens of amino acids) is rather heterogeneous and with multiple competing pathways (Dill, 2012; Jackson, 1991; Schönfelder, 2016; Rhoades, 2004). Conversely, larger proteins have a more complex folding kinetics and relatively more defined folding mechanisms (Bhatia, 2024; Cecconi, 2005; Dingfelder, 2021). In particular, chains consisting of a few hundred amino acids typically fold by docking partially folded substructures, called foldons (Ianeselli, 2018; Paci, 1999). The emergence of a prominent folding mechanism is arguably further enhanced in vivo by the vectorial nature of ribosomal co-translational folding and because the folding rate-limiting step (i.e., the addition of amino acids to the nascent chain) is much slower than the typical refolding rates (Ingolia, 2014). Such a perspective not only conceptually justifies developing reproducible models for the folding pathways of specific proteins but also suggests that, if

[1]Department of Cellular, Computational and Integrative Biology (CIBIO), University of Trento, Via Sommarive 9, Trento 38121, Italy. [2]Sibylla Biotech, Via Lillo del Duca 10, Bresso 20091, Italy. [3]Department of Biomedical Sciences, University of Padova, Viale G. Colombo 3, Padova 35121, Italy. [4]Bicocca Quantum Technology Center & Physics Department, University of Milan Bicocca, Piazza della Scienza 2/A, Milan 20126, Italy. [5]CIMUS Biomedical Research Institute and Department of Medical Sciences, University of Santiago de Compostela-IDIS, Santiago de Compostela 15782, Spain. [6]Veneto Institute of Molecular Medicine (VIMM), via Orus 2, Padova 35129, Italy. [7]Department of Biomedical Sciences, University of Padova, Via Ugo Bassi 58/B, Padova 35131, Italy. [8]Centre for Agriculture Food Environment, University of Trento, Trento, Italy. [9]Research and Innovation Centre, Fondazione Edmund Mach, San Michele All'Adige, TN 38010, Italy. [10]International School for Advanced Studies (SISSA), Via Bonomea 265, Trieste 34136, Italy. [11]INFN-TIFPA, Via Sommarive 14, Trento 38123, Italy. [12]These authors contributed equally: Dino Gasparotto, Annarita Zanon, Valerio Bonaldo. ✉E-mail: pietro.faccioli@unimb.it; emiliano.biasini@unitn.it

co-translational folding does take place through relatively consistent steps, kinetically relevant intermediate conformations could influence the whole protein folding process. Protein folding mechanisms and their defects have been linked to protein homeostasis (or proteostasis) and disease at multiple levels (Wilkinson, 2010). Indeed, proper folding of newly synthesized proteins is fundamental for maintaining biological functions. The process is closely monitored by various cellular pathways, which intervene to correct aberrantly folded species or reroute them to degradation (Sun, 2019). Sufficiently long-lived off-pathway conformational states that are spontaneously visited during the folding process may be recognized by the quality control machinery as improperly folded species, thus contributing to modulate protein homeostasis. In contrast, the possibility that on-pathway transient conformational states might hold intrinsic biological functions and be directly implicated in regulatory mechanisms has remained largely unexplored (Biasini, 2025).

Post-translational modifications (PTMs) are chemical changes that occur to proteins after they have been synthesized, significantly altering a protein's function, stability, localization, and interactions, playing a crucial role in regulating cellular processes (Zhong et al, 2023). PTMs include various modifications such as phosphorylation, ubiquitination, glycosylation, acetylation, sumoylation, and methylation. Among these, phosphorylation is one of the most common and important. It involves adding a phosphate group to the side chain of specific amino acids, serine, threonine, or tyrosine residues. Kinases are the enzymes catalyzing such additions of phosphate groups, whereas phosphatases catalyze their removal. Phosphorylation plays diverse roles in cellular regulation. It acts as a molecular switch in signaling pathways, activating or deactivating proteins that deliver signals within the cell. It is also crucial for cell growth, division, and response to external stimuli. In addition, phosphorylation can alter the activity of enzymes, enhancing or inhibiting their function and allowing cells to adapt to environmental changes rapidly. Adding a phosphate group can also create or disrupt binding sites for other proteins, influencing complex formation and cellular pathways. Furthermore, phosphorylation can change the localization of proteins within the cell, directing them to specific compartments where they exert their function (Walsh, 2005). Lastly, phosphorylation can mark proteins for degradation or protect them from it, thereby controlling protein turnover and maintaining protein homeostasis (Hunter, 1998). Through these roles, phosphorylation is a critical mechanism for dynamic protein regulation, influencing almost every aspect of cell biology.

In this manuscript, we combine experiments, theory, and simulations to investigate whether phosphorylation at sites predicted to be buried in the native protein structure can occur co-translationally and influence protein expression and stability. Our analyses established that, differently from any other PTMs, phosphorylation may involve amino acids locked into rigid domains buried inside native protein cores. Computer simulations and experimental correlates show that these cryptic phosphosites are transiently exposed in long-lived protein folding intermediates and destabilize native structures when phosphorylated, leading to the rapid degradation of polypeptides. Our findings introduce a novel perspective on the regulation of protein expression, suggesting that co-translational phosphorylation events occurring on cryptic residues may determine the folding and stability of

polypeptides. This concept expands our understanding of protein homeostasis with significant implications for physiological processes and therapeutic strategies.

## Results

### Bioinformatic analysis to mine cryptic phosphosites in the human proteome

By definition, PTMs occur on proteins that have completed their folding pathway and reached their native states. However, there are cases in which the modifying enzyme intervenes on the target protein while the folding process is still ongoing. For instance, glycosylation can occur co-translationally as soon as the target polypeptide emerges from the ribosomal channel (Ruiz-Canada et al, 2009). Such chemical modifications likely involve intermediates that transiently appear along the folding process. This conclusion suggests that specific PTMs may affect the overall expression of a polypeptide by influencing its folding pathway. We sought to explore this possibility for all the main PTMs in the human proteome, including phosphorylation, ubiquitination, acetylation, methylation, sumoylation, in addition to glycosylation. PTMs require the target amino acids to be exposed on the protein surface, as the enzymes responsible for adding or removing the specific chemical groups need direct access to the side chains. Conversely, an amino acid buried within the protein's core is shielded from the solvent and becomes inaccessible for enzymatic modification. We hypothesized that PTMs influencing protein folding pathways may involve residues exposed in on-pathway protein folding intermediates while being hidden from the solvent once the protein has reached is native conformation. To test this hypothesis, we built a computational pipeline aimed at systematically identifying PTMs in the human proteome occurring on hidden residues, named cryptic sites (Fig. 1). First, data extracted from the database PhosphoSitePlus, which includes the largest collection of experimentally validated PTMs, were cross-referenced with the DeepMind's AlphaFold protein structure database. This step assigned each PTM from PhosphoSitePlus to the corresponding protein structure in AlphaFold. In an attempt to significantly increase the reliability of our dataset, we restricted the analysis to PTMs lying within structured domains showing an AlphaFold confidence score (pLDDT) above 65. We then assessed the RSA value of the residues corresponding to the different PTMs in order to determine solvent exposure in the native state (Fig. 2). PTM residues with an RSA value below 0.15 were classified as cryptic, reflecting their poor solvent exposure. The analysis revealed that ubiquitination, acetylation, methylation, and sumoylation share a normal-like distribution centered around values of RSA between 0.4 and 0.8, indicating that the vast majority of these PTMs occur on residues exposed on the surface of the corresponding proteins. As expected, the distributions of glycosylation differed significantly, featuring a bimodal distribution with two peaks centered around RSA values between 0.15 and 0.3 and between 0.4 and 0.6. Such bimodality reflects the aforementioned propensity of glycosylation to occur both co- and post-translationally. In contrast to the distributions of all other PTMs, we observed a complex multimodal distribution for phosphorylation. This PTM presented a large density of entries with RSA values ranging between 0.2 and 0.6, but

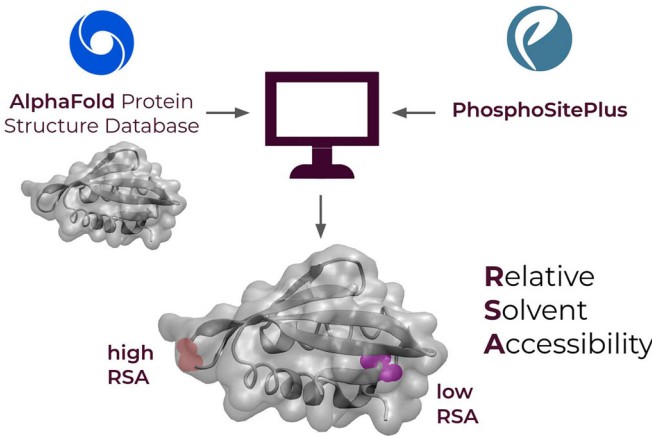

**Figure 1.  Schematic illustration of the workflow adopted to identify cryptic PTMs.**

Data from PhosphoSitePlus, which holds an extensive collection of experimentally validated PTMs, were cross-referenced with protein structures from DeepMind's AlphaFold database. This approach matched each PTM entry from PhosphoSitePlus to its corresponding protein structure in AlphaFold. The analysis focused on PTMs within structured regions, specifically those with an AlphaFold confidence score (pLDDT) exceeding 65. RSA values of PTM-modified residues were calculated to assess their solvent exposure in the native state. Residues with RSA values below 0.15 were categorized as cryptic, indicating minimal solvent exposure.

also a sharp peak between 0 and 0.15, likely indicative of residues buried into protein cores. These cryptic phosphosites accounted for 26.4% of all plotted phosphorylated residues. Subsequent stratification of the new dataset of cryptic phosphosites highlighted differences in the relative distribution of the three phosphorylation-permissive amino acids (i.e., serine, threonine, and tyrosine), with tyrosine presenting a larger proportion of cryptic phosphosites (Fig. EV1). This observation is consistent with the differential characteristics of phosphotyrosine versus phospho-serine/threonine sites, with the former being predominantly located within structured protein domains, whereas the latter sites can be found throughout whole protein sequences, often including unstructured domains. Moreover, RSA values of cryptic phosphosites did not correlate with proteins' length (Fig. EV2). Approximately 65% of proteins containing cryptic phosphosites presented only 1 or 2 of such residues, while only less than 10% showed 5 or more sites (Fig. EV3). Overall, these results unexpectedly revealed that a substantial number of phosphorylated amino acids in the human proteome are apparently cryptic, lying within the inner core of proteins' native states.

## Dynamic and structural filtering of cryptic phosphosites

The RSA values of cryptic phosphosites were computed on static protein structures predicted by AlphaFold without considering any thermal fluctuations. However, the latter can activate concerted

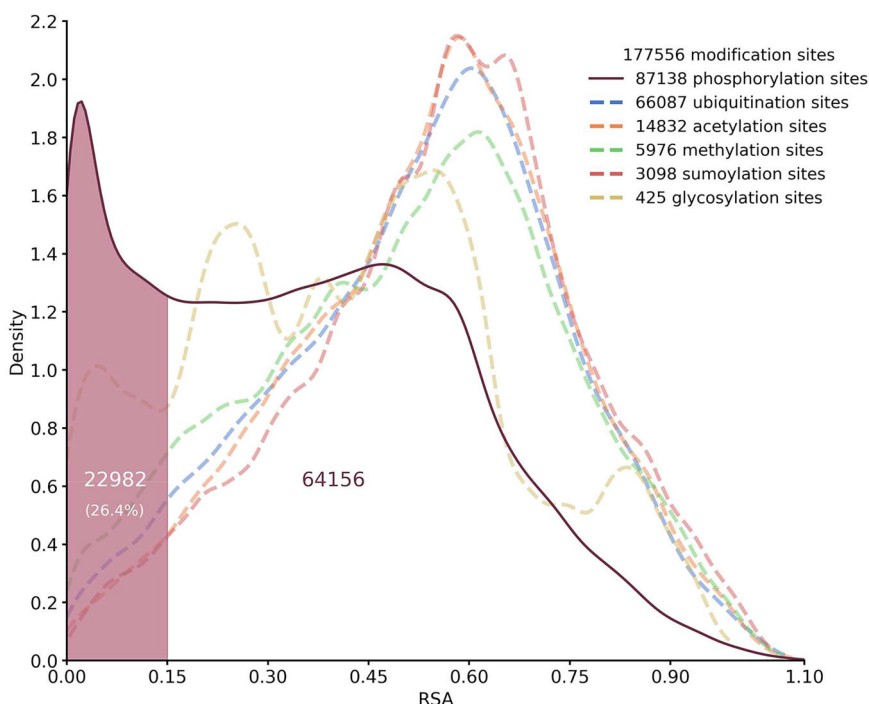

**Figure 2.  Graphic distribution of the occurrence of different PTMs in relation to the RSA of the corresponding target amino acid.**

The figure illustrates the relative distribution of distinct PTMs across a range of RSA values, representing solvent accessibility of the target residues in their native protein structure. Each PTM category is plotted against its corresponding RSA value, highlighting patterns of distribution and solvent exposure. The analysis reveals differences in solvent accessibility between PTM types, with cryptic modifications clustering at low (<0.15) RSA values, indicative of buried residues. The distribution provides insights into the structural environments favored by different PTMs, underlining the unique behavior of phosphosites compared to other modifications.

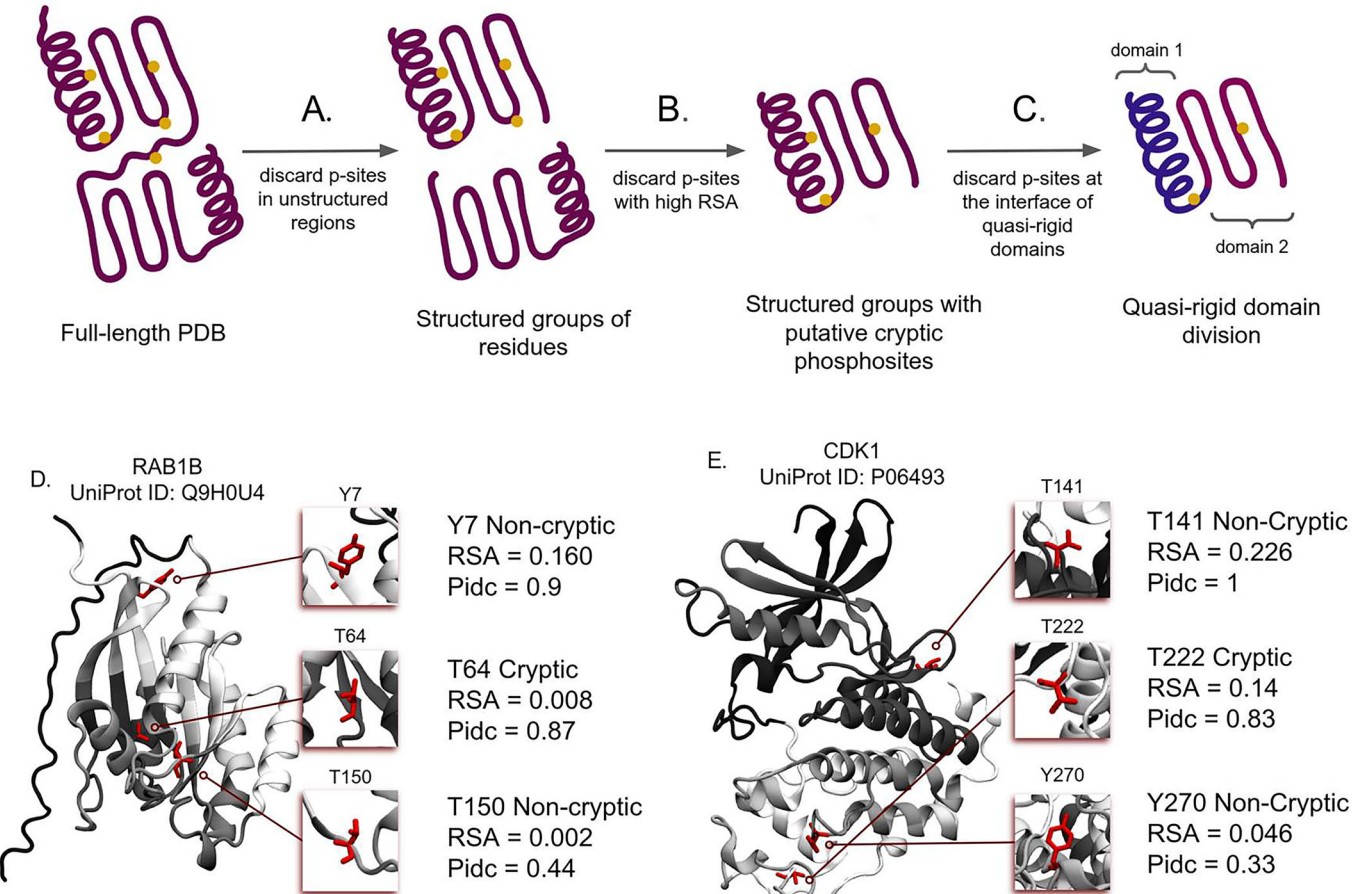

**Figure 3. Schematic illustration of the workflow adopted for the dynamic and structural filtering of post-translationally exposed sites.**

(A) Low-confidence segments, such as loops, linkers, or disordered regions with AlphaFold scores below 65, were removed from the starting protein dataset. The refined dataset included either single protein domains or groups of contiguous domains treated as single entries if at least five residues were within 5 Å of one another. Cryptic phosphosites located in domains with fewer than 40 residues were excluded. (B) The RSA of cryptic phosphosites was recalculated for the updated entries, excluding residues that presented an RSA above 0.15. (C) SPECTRUS identified quasi-rigid domains by creating β-Gaussian models for each entry. Cryptic phosphosites located in quasi-rigid domains of the highest SPECTRUS quality scoring group were further filtered, excluding those whose side chains make fewer than 80% of intra-domain contacts. (D, E) Example of two proteins containing true and false cryptic phosphosites. The structures of RAB1B ((D), UniProt ID: Q9H0U4) and CDK1 ((E), UniProt ID: P06493) are depicted in new cartoon style. The color of the image does not reflect any structural property but instead it is used to distinguish different quasi-rigid domains. In particular, black regions identify unstructured domains, whereas shadows from dark gray to white identify quasi-rigid domains. Boxes highlight the individual phosphosites and the related amino acid position, cryptic or non-cryptic classification, post-processing RSA value, and proportion of intra-domain contacts (Pidc).

large-scale displacements of groups of amino acids (Micheletti, 2013), termed quasi-rigid or dynamical domains, whose relative motion may change the solvent exposure of putative cryptic sites, leading to higher RSA values. Therefore, our compiled database of cryptic phosphosites likely contains a relevant amount of post-translationally exposed sites that can become transiently exposed during the thermally activated internal dynamics of the protein. To address this issue, we used SPECTRUS, a recently described computational method to identify dynamical domains in proteins (McGibbon, 2015). In particular, we designed a multi-step filtering procedure based on structural and dynamic analyses, aimed at excluding cryptic phosphosites that become exposed to the solvent as a result of thermal fluctuations (Fig. 3). The first step involved removing low-confidence segments (such as loops, linkers, or short disordered regions) from the initial protein dataset (Fig. 3A). This refined dataset included individual or contiguous protein domains,

with entries containing at least five residues within 5 Å of each other. Cryptic phosphosites within domains smaller than 40 residues were excluded due to the low reliability of SPECTRUS in handling short polypeptides. The RSA of cryptic phosphosites was recalculated, filtering out residues with an RSA above 0.15 (Fig. 3B). Finally, SPECTRUS identified quasi-rigid domains in this updated protein dataset using β-Gaussian models (Fig. 3C). Cryptic phosphosites in these domains with top SPECTRUS quality scores were excluded if their side chains had fewer than 80% intra-domain contacts (examples of true and false cryptic phosphosites are illustrated in Fig. 3D,E). Such highly stringent filtering steps generated a final database of 10,606 cryptic T, S and Y phosphosites out of ~218,000 total phosphosites, belonging to 5496 different proteins out of ~18,000 total phosphorylated proteins annotated in PhosphoSitePlus (Dataset EV1; Fig. EV4). These data indicate that ~5% of all known phosphosites are cryptic. The number translates

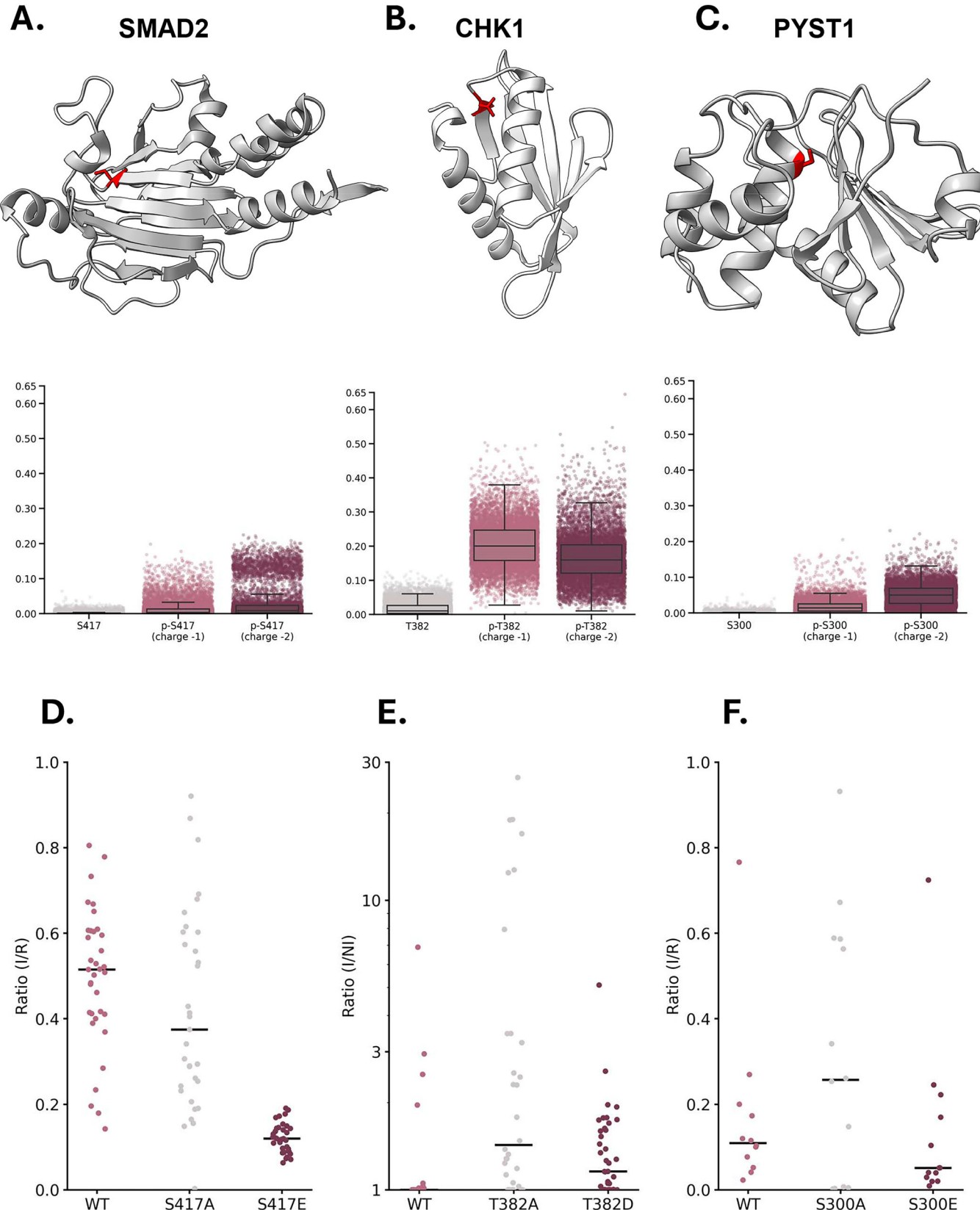

**Figure 4.  Experimental characterization of the effect of phosphoablative and phosphomimetic mutations in SMAD2, CHK1, and Pyst1.**

(A–C) Illustration of the three-dimensional structure of SMAD2 (**A**), the KA1 domain of CHK1 (**B**), and Pyst1 (**C**). The side chain of each individual cryptic phosphosite is shown in a different color. The graphs below each structure report the graphical plot of RSA values calculated for different amino acidic variants at the related position (from left to right): for SMAD2, unphosphorylated S417, phosphorylated S417 with charge −1, phosphorylated S417 with charge −2; for CHK1, unphosphorylated T382, phosphorylated T382 with charge −1, phosphorylated T382 with charge −2; for Pyst1, unphosphorylated S300, phosphorylated S300 with charge −1, phosphorylated S300 with charge −2. RSA values were calculated along a 1 μs MD simulation. (**D, E, F**) Dot plots reporting the relative expression levels of clones for each individual construct: for SMAD2, S417 (WT), S417A (phosphoablative mutant), and S417E (phosphomimetic mutant); for CHK1, T382 (WT), T382A (phosphoablative mutant), and T382D (phosphomimetic mutant); for PYST1, S300 (WT), S300A (phosphoablative mutant), and S300E (phosphomimetic mutant). All the protein variants were tagged with the FLAG epitope to facilitate quantification. The expression of each clone was assessed by western blotting using an anti-FLAG antibody. For SMAD2 and PYST1, protein levels in the different clones were expressed as the ratio of corresponding protein lanes in the induced samples (I) and a reference control (R) to allow cross-comparison among different gels. For CHK1, we observed leakage in the non-induced controls of expressing clones. Therefore, each signal was expressed as the ratio between induced (I) and non-induced (NI) signals. Samples showing a negative ratio were set to 1. Statistical differences were evaluated by a multiple Mann–Whitney $U$ test: for SMAD2, WT vs S417A, $P = 0.11626$, WT vs S417E, $P = 0.00001$ (****), S417A vs S417E, $P = 0.00001$ (****); for CHK1, WT vs T382A, $P = 0.14486$, WT vs T382D, $P = 0.08444$, T382A vs T382D, $P = 0.00555$ (*); for PYST1, WT vs S300A, $P = 0.58915$, WT vs S300E, $P = 0.16544$, S300A vs S300E, $P = 0.48166$.

to ~33% of phosphorylated proteins in the human proteome presenting at least one cryptic phosphosite.

## Computational and experimental validation of cryptic phosphosites in model proteins

In order to validate the results of the computational analyses, we selected cryptic phosphosites for further computational and experimental assessment. The selection was limited to phosphosites occurring on protein domains shorter than 200 amino acids to facilitate in silico analyses. Furthermore, we selected proteins that contain cryptic sites phosphorylated by identified kinases. Two candidates out of such selection included phosphosites S417 of the SMAD family member 2 (SMAD2, PDB 1DEV), a protein involved in signal transduction (Miyazawa et al, 2024), and residue T382 lying within the kinase associated-1 (KA1) domain of the checkpoint kinase 1 (CHK1, PDB 5WI2), a factor involved in DNA damage response and cell cycle checkpoint regulation (Patil et al, 2013). Importantly, in both cases, the phosphorylation of the cryptic residues had been previously reported to be associated with loss of function effects or proteasomal degradation (Yan et al, 2012; Gong et al, 2018). One additional phosphosite, the S300 of PYST1 (PDB 1MKP), a member of the dual specificity protein phosphatase subfamily, which was removed by our dynamic filtering, was added to the list as a negative control (Fig. 4) (Seternes et al, 2019). To explore the impact of phosphorylation of the three selected residues, we performed 1 μs MD simulations imposing either −1 or −2 as a possible charged state of the phosphate group. We then compared the RSA of the unphosphorylated forms with their phosphorylated counterparts. Not surprisingly, in all three cases we observed that phosphorylation substantially increased RSA of the corresponding residue (Fig. 4A–C). The rise in solvent accessibility is likely due to the charges from the phosphate group and the steric hindrance it introduces, both of which prompt a conformational shift in the proteins, thereby increasing the exposure of the residues to the solvent. To experimentally investigate the effect of phosphorylation on protein expression, we performed cellular clonal analyses. First, we engineered each protein in three different forms: a WT version carrying the original residue at the corresponding cryptic position and phosphoablative or phospho-mimetic mutants. The formers were designed to substitute the original amino acid with an alanine, thus abrogating the possibility of being phosphorylated. Conversely, glutamate or aspartate was

introduced to mimic the spatial hindrance and electrostatic properties of phosphoserine and phosphothreonine, respectively. A FLAG-tag was also added to the C-terminus of each protein variant to facilitate and standardize immunodetection. Each construct was then inserted into an expression vector downstream of a Tet-inducible promoter, which allows controlling protein expression through the addition of doxycycline. Finally, plasmids encoding for the different protein versions were stably transfected into HEK293 cells, and protein levels were evaluated by western blotting upon 24–48 h induction. Clones that exhibited an expression level above a defined threshold were counted as positive (Figs. 4D–F, EV5A–C, EV6A–C, and EV7A–C). We observed small differences in the basal expression efficiency of SMAD2, CHK1, and Pyst1 in their WT forms (percentages of positive clones were: SMAD2-S417, 97%; CHK1-T382, 100%; Pyst1-S300, 92%). The expression efficiency of the phosphoablative mutants did not significantly differ from the corresponding values of the WT forms (SMAD2-S417A, 94%; CHK1-T382A, 92%; Pyst1-S300A, 64%). Conversely, phosphomimetic mutants for SMAD2 and CHK1, but not for Pyst1, showed a drastically reduced number of positive clones (SMAD2-S417E, 29%; CHK1-T382D, 49%; Pyst1-S300E, 69%). These results suggest that the phosphorylation of cryptic sites in SMAD2 and CHK1 can directly influence their expression efficiency.

To further corroborate these observations, we sought to compare the clearance rate of the different protein variants. We conducted these analyses on selected CHK1 clones. Following the induction of the three different CHK1 variants for 48 h, protein synthesis was arrested by removing doxycycline and adding the ribosomal inhibitor cycloheximide (CHX). This method enabled us to track protein loads at defined time points after stopping protein synthesis, allowing the construction of a kinetic turnover profile from which we extrapolated a half-time value for the different CHK1 forms (Fig. 5A–C). The measurements revealed a severe reduction of the turnover time for the phosphomimetic CHK1 mutant as compared to the WT or the phosphoablative counter-parts (half-time in hours were: CHK1-T382, 6.4 ± 1.7; CHK1-T382A 7.4 ± 1.9, CHK1-T382D, 1 ± 0.5; see also Fig. EV8A).

Finally, we used a previously described enhanced path sampling approach to perform all-atom simulations of the folding process of the CHK1-KA1 domain. Intriguingly, this physics-based recon-struction revealed the existence of three kinetically relevant, metastable intermediates that expose the cryptic phosphosite

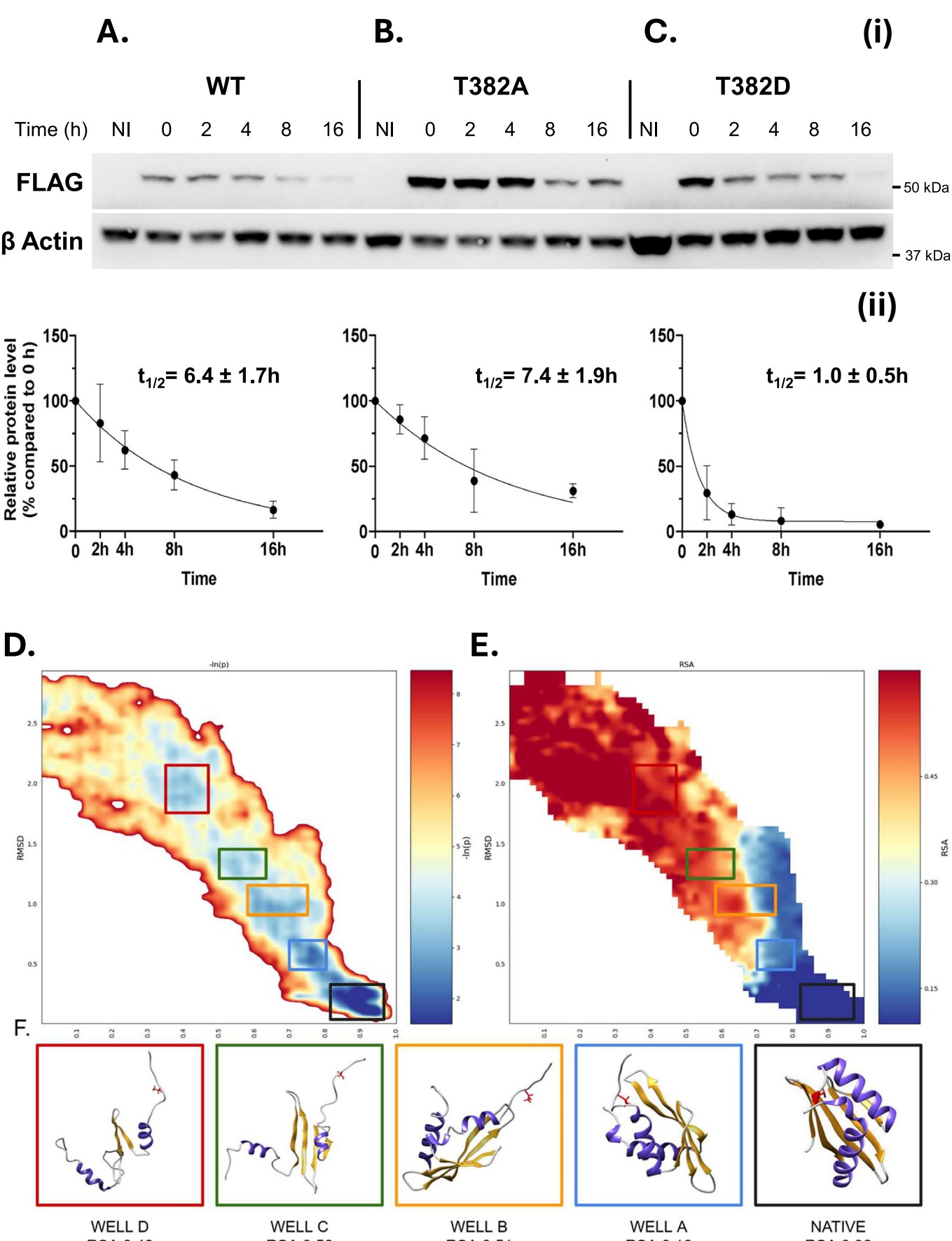

**Figure 5. Experimental and computational investigation of cryptic phosphosite T382 in CHK1.**

(A–C) CHK1 stability was analyzed in three selected clones expressing CHK1 variants at codon T382: WT T382 (A), phosphoablative mutant T382A (B), and phosphomimetic mutant T382D (C). After doxycycline induction (72 h, 1 μg/mL), protein synthesis was stopped with cycloheximide (CHX), and CHK1 levels were monitored at 0, 2, 4, 8, and 16 h using western blotting with anti-FLAG antibody; β-actin signal served as loading control. For each time point, ≥3 biological replicates were produced (except for the T382D clone at 16 h, where only one replicate was obtained). Panels (i) display representative western blots for each CHK1 form, detected with an anti-FLAG antibody. Panel (ii) illustrates the quantified data (mean ± standard deviation), normalized to the baseline level ($t_0$). The degradation kinetics were modeled using a one-phase decay curve from which half-time ($t_{1/2}$) values were derived. Data analysis and curve fitting were performed using the GraphPad Prism software. (D) Atomistic reconstruction of the CHK1 KA1 domain folding pathway. Lower-bound approximation of the transition path energy is related to the folding of KA1. The energy is plotted as the negative logarithm of the probability distribution (-ln(p)), which is expressed as a function of the collective variables Q and RMSD (65 × 65 bin matrix). A Gaussian blur was applied. The highly populated native state appears as expected in the bottom-right corner (high Q and low RMSD, black rectangle). The indexed squares define the most-populated partially unfolded regions of interest (-ln(p)). Well thresholds from most to least stable: (A) ≤2.5 k. bT, (B) ≤3.5 kbT, (C) ≤3.5 kbT, (D) ≤3 kbT. (E) Distribution of RSA of amino acid T382 along the transition path. The highest values are associated with the unfolded state (top left), while in the native state, the residue is consistently below 0.15. (F) Representative conformations for each cluster. All clusters are explored at least by 2/9 LB conformations. Residue T382 is displayed in red, α-helices are colored in purple, β-sheets in orange.

T382 to the solvent (Fig. 5D–F). Similarly, also for SMAD2 we found kinetically relevant folding intermediates exposing S417 to the solvent (Fig. EV8B–D).

Collectively, these results provide evidence for a regulatory mechanism by which the phosphorylation at cryptic phosphosites can occur during the folding process and affect protein homeostasis by drastically increasing the degradation rate and leading to rapid turnover.

## Phylogenetic analysis of human cryptic phosphorylation sites

Amino acids lying within the inner core of proteins are generally more conserved than those on the surface. The conservation stems from the critical role these residues play in maintaining the protein's structural stability. Core residues are often involved in hydrophobic interactions that help stabilize the protein's three-dimensional structure, making any variation at these sites more likely to disrupt the whole protein's architecture. By contrast, surface residues tend to be more tolerant to mutations because their contribution in stabilizing the proteins' native conformation is generally lower (Cagiada et al, 2023). A different pattern of evolutionary conservation has been observed across diverse proteins, where mutations in core residues often negatively affect protein expression, while variation of surface residues can alter proteins' function and range of interactions without compromising the protein's overall stability (Faure and Koonin, 2015). Cryptic phosphosites are defined by their low RSA, a property shared with amino acids lying within the inner core of proteins. Thus, we sought to test the hypothesis that the cryptic phosphosites listed in our dataset are more evolutionary conserved than non-cryptic phosphosites. To test this hypothesis, we selected 137 representative and non-redundant eukaryotic organisms with a reference proteome deposited in uniprot.org (Dataset EV2). Among these proteomes, we defined orthogroups by the software OrthoFinder. An orthogroup is a set of genes from different species that originated from a single gene in the last common ancestor of those species. These genes are called orthologs and usually retain similar functions across species, although they can undergo divergence over time. Orthogroups provide a framework to analyze gene evolution across species, as they include all the orthologous genes derived from the same ancestral gene. In comparative genomics, orthogroups are essential for studying functional conservation and

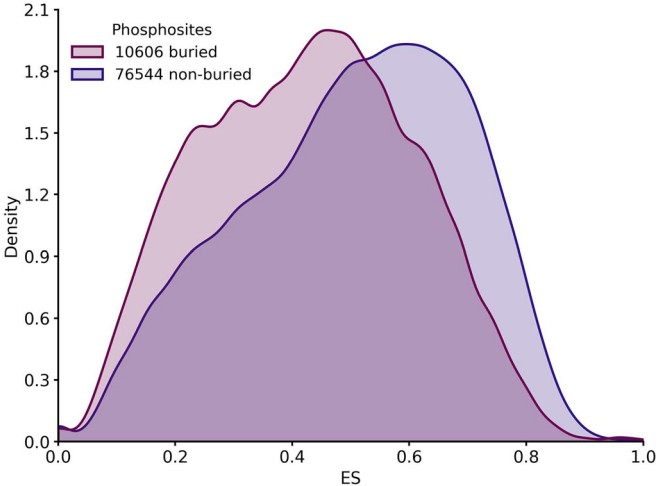

**Figure 6. Graphic distribution of evolutionary conservation of cryptic and non-cryptic phosphosites.**

The KDE plot reports the distribution of ES for cryptic (colored in light purple) and non-cryptic (colored in dark purple) phosphosites.

evolutionary relationships between genes across different organisms. We performed a multiple sequence alignment only for orthogroups containing at least a human gene. Finally, we cross-referenced datasets of cryptic and non-cryptic phosphosites with the alignment of the corresponding orthogroups and calculated the entropic score (ES) for each site. ES is a metric used to express the randomness associated with a particular system, such as the evolutionary conservation of amino acids. We observed two distinct distributions for cryptic and non-cryptic phosphosites, with the former centered on lower ES values (median ES = 0.43) as compared to the latter (median ES = 0.52) (Fig. 6). Due to the large sample sizes in our analysis, statistical tests inevitably yield very low *P* values, even when differences in mean are minimal and the standard deviations relatively small. To better assess the practical relevance of these differences, we calculated Cohen's *d* as a measure of effect size. The comparison between cryptic and non-cryptic phosphosites yielded an effect size (Cohen's *d* = 0.4028) slightly lower than the one obtained for residues lying within protein cores or exposed on protein surfaces (Cohen's *d* = 0.5126), both indicating a modest but meaningful shift in entropic scores. In

contrast, the comparisons between cryptic phosphosites and all core residues, as well as non-cryptic phosphosites and all surface residues, showed negligible effect sizes (Cohen's $d = 0.0245$ and 0.1326, respectively). These findings indicate that cryptic phosphosites tend to be more evolutionarily conserved across eukaryotes than non-cryptic phosphosites. Importantly, the distributions of cryptic and non-cryptic phosphosites did not differ from those of all cryptic and non-cryptic residues in terms of effect size (Fig. EV9). These results confirm that cryptic phosphosites share a higher evolutionary conservation with amino acids lying within the inner core of proteins.

### Several cancer-associated somatic mutations occur in cryptic phosphosites

Our data suggest that the phosphorylation of cryptic sites may influence protein expression, leading us to hypothesize that mutations in certain cryptic phosphosites could be associated with human pathologies. To explore this possibility, we conducted a cross-referencing analysis searching for cryptic phosphosites in two comprehensive datasets: COSMIC (Catalog Of Somatic Mutations In Cancer), which documents somatic mutations identified in human cancers, and PTMVar, a database listing disease-associated missense mutations occurring in post-translationally modified residues curated from PhosphoSitePlus. First, we compared the proportion of cryptic or non-cryptic phosphosites associated with cancer- and disease-related mutations in each group from the COSMIC and PTMVar datasets. The percentage of phosphosites associated with the two repositories is the same for cryptic and non-cryptic sites (Appendix Table S1). This observation suggests that, despite their different structural and regulatory features, both site types occur similarly in disease contexts. Next, we found that 204 mutations in COSMIC are associated with cryptic phosphosites that have been annotated as cancer-related (Dataset EV3) and 138 mutations in PTMVar linked to cancer and other human pathologies (Dataset EV4). Notably, among these entries, we identified 11 phosphomimetic substitutions that occurred in hotspot loci of tumor-suppressor genes, including the Von Hippel–Lindau tumor suppressor (pVHL), the tumor protein 53 (p53), and the phosphatase and tensin homolog (PTEN) (Appendix Table S2). Interestingly, phosphorylation of cryptic serine residues in PTEN has been reported to reduce its stability and membrane localization (Vazquez et al, 2000). These findings suggest that phosphomimetic mutations occurring in cryptic phosphosites may directly contribute to increased tumor cells' fitness by down-regulating the expression or altering the function of specific tumor-suppressor proteins.

## Discussion

In this work, we comprehensively analyzed the solvent accessibility of post-translationally modified residues across the human proteome. We unexpectedly found that, in contrast to other PTMs, phosphorylation can occur on residues buried within inner regions of native protein structures. Theoretical modeling of the native structural dynamics revealed that a significant fraction of the buried residues lie within the core of rigid protein sub-domains, thus are unlikely to become solvent exposed under the effects of

spontaneous thermal fluctuations or protein-protein interactions. Experimental validation in model proteins confirmed that phosphomimetic substitutions at these sites decrease protein expression efficiency and drastically reduce protein half-lives. All-atom, physics-based simulations of the protein folding process performed by means of a dedicated enhanced sampling algorithm show that, at least in the representative cases we considered, the phosphorylation sites are solvent exposed exclusively in metastable folding intermediates transiently visited along productive folding routes. Furthermore, phylogenetic analyses indicate, in comparison to non-cryptic amino acids, cryptic phosphosites possess a higher evolutionarily conservation typical of residues lying within protein cores. Finally, we identified cases in which phosphomimetic mutations in human cancers at cryptic phosphosites involve specific tumor suppressors, directly indicating a mechanistic significance for this phenomenon. Overall, our findings introduce an entirely new perspective on protein expression regulation with profound biological and therapeutic implications by suggesting that the folding process of several polypeptides could be determined by phosphorylation of cryptic sites.

### Dataset reliability

The reliability of our dataset is strongly corroborated by multiple lines of evidence. First, unlike other PTMs, which we found almost exclusively occurring on solvent-exposed residues, phosphorylation uniquely involves a distinct subset of amino acids showing extremely low solvent accessibility. This unique feature reinforces the concept that phosphorylation at cryptic sites follows a different accessibility mechanism, limited to folding states where these sites are temporarily exposed, such as during protein folding. The existence of cryptic phosphosites also aligns well with previous observations made in smaller datasets (Jiménez et al, 2007). Second, we filtered our dataset of cryptic phosphosites with further computational methods. Dynamical refinement steps were integrated to remove potential post-translationally exposed sites, thus focusing only on PTMs lying within rigid protein domains. Such a filtering step narrows our analysis to a specific subset of potential regulatory sites. Despite such a stringent filtering procedure, we cannot completely rule out the possibility that some cryptic sites may become accessible in particular conditions from natively folded proteins, for example as a result of conformational changes mediated by the specific interaction with other proteins. However, validation of some of our dataset's entries through experimental assessments on selected proteins supported the predictions of our computational analyses. Finally, in contrast to solvent-exposed phosphoresidues, cryptic phosphosites show a higher evolutionary conservation typical of amino acids lying within protein cores. This feature likely reflects the evolutionary constraint on hydrophobic core residues, which tend to be more conserved due to their role in structural stability. Of note, phosphotyrosine is known to have distinct evolutionary trajectories as compared to phosphoserine and phosphothreonine. Indeed, the latter signaling pathways are ancient and broadly conserved across eukaryotes, while phospho-tyrosine signaling is thought to have emerged later, around the origin of the metazoan lineage, in association with the expansion of tyrosine kinases and SH2-domain–containing proteins. Such a divergence has important implications for interpreting the evolutionary conservation of cryptic phosphosites, particularly tyrosine

residues, and may explain differences in selective pressures and functional relevance across lineages. The high conservation does not imply that phosphorylation at cryptic sites is functionally selected across species. Instead, when such residues are phosphorylated, as we observe in the human proteome, the effect is often destabilizing and associated with protein degradation. Thus, our analysis does not establish that the phosphorylation of cryptic residues is conserved across species, only that the residues themselves are highly conserved. Indeed, the high degree of evolutionary conservation confirms the hidden nature of cryptic phosphosites and suggests that these residues may play important roles across biological systems. Collectively, these findings underscore the robustness and reliability of our dataset, thus providing a completely new perspective on the biological regulation of thousands of phosphorylated proteins.

## Insights into SMAD2 and CHK1 biology

Our investigation into SMAD2 and CHK1 biology provides examples of the possible impact of cryptic phosphorylation on protein regulation. SMAD2 is a signaling protein involved in the TGF-β pathway, playing a key role in regulating cell growth, differentiation, and development by transmitting signals from the cell surface to the nucleus (Miyazawa et al, 2024). Interestingly, S417 phosphorylation has already been linked to decreased SMAD2 activity, impacting the downstream TGF-β signaling pathway (Yan et al, 2012). Our identification of S417 as a cryptic phosphosite is particularly intriguing as it suggests that this residue may function as a regulatory switch of SMAD2 expression, directly influencing TGF-β signaling. This conclusion implies that the S417 could be a unique site for therapeutic interventions aimed at modulating SMAD2 activity in diseases associated with abnormal TGF-β signaling, such as cancer and fibrosis. We also found that CHK1, a kinase involved in cell cycle regulation and DNA damage response, is regulated through phosphorylation at the cryptic sites T382. This observation aligns well with existing literature on CHK1's regulation through phosphorylation at multiple sites (Gong et al, 2018), although our data suggest that phosphorylation of T382 is sufficient for re-routing the protein to degradation. Finally, computational simulations provide evidence that both S417 in SMAD2 and T382 in CHK1 are exposed to the solvent in specific intermediate states populated along their folding pathways. These findings directly support a model by which the expression of SMAD2 and CHK1 may be dynamically regulated during their folding pathway by phosphorylation at the corresponding cryptic phosphosites.

## Phosphorylation as a co-translational event

Our research sheds light on a potential mechanism whereby phosphorylation at cryptic sites may occur as a co-translational event, adding a fresh perspective to our understanding of protein chemical modifications. Typically, phosphorylation requires exposed residues on the surface of natively folded proteins. Cryptic phosphosites provide a radically different paradigm where residues are modified while proteins are still in the folding process. Such a transient accessibility during folding exposes cryptic residues to kinases with specific spatial and kinetic constraints dictated by the folding process itself. In this framework, other factors like

chaperons participating in the folding pathway may play fundamental roles in regulating the accessibility of cryptic sites to the modifying enzymes. Indeed, the existence of ribosome-associated kinases and phosphatases suggests the possibility that these enzymes may directly modify nascent polypeptide chains during translation (Simsek et al, 2017). This conclusion supports the notion that specific cryptic phosphosites could be part of a co-translational protein regulation machinery. Interestingly, phosphorylation at cryptic sites could produce effects that go beyond the simple negative regulation of protein expression. While the cryptic phosphorylation cases investigated in this manuscript are primarily associated with destabilizing effects, leading to increased protein degradation, we cannot exclude that in several cases cryptic phosphorylation could positively affect protein folding, stability, or reroute the folding pathway toward alternative conformations. For example, phosphorylation of a cryptic site could favor recruitment of chaperons or phosphorylation reading modules expanding the conformational landscape of the protein, as already observed for the cochaperone Cdc37 (Bachman et al, 2018). Overall, these conclusions support a previously unappreciated role for phosphorylation in the co-translational fine-tuning of protein expression and stability.

## Relevance of cryptic phosphosites in physiology and therapy

It is appealing to hypothesize that cryptic phosphorylation may serve as an intrinsic quality control mechanism to prevent the accumulation of misfolded proteins, acting as a safeguard in the protein folding process. This regulatory strategy could selectively tag proteins for degradation before they achieve their mature conformation, ensuring that defective or improperly folded proteins do not accumulate and disrupt cellular homeostasis. In this scenario, cryptic phosphosites could thus represent critical kinetic control points within the folding pathway. During the normal folding process, cryptic phosphosites may remain only briefly exposed by folding intermediates, escaping modification by kinases. However, in the case of incorrect or stalled folding, folding intermediates may linger in their partially folded states, prolonging the exposure of cryptic phosphosites. This extended availability would allow kinases to phosphorylate the exposed residues, effectively marking the protein as defective. The phosphorylation of cryptic sites in improperly folded polypeptides could thus serve as a co-translational signal for cellular degradation pathways such as the ubiquitin-proteasome system or autophagy. This process would ensure that faulty proteins are efficiently cleared before aggregating or interfering with cellular functions. Importantly, this hypothesis introduces a kinetic dimension to protein quality control, where the timing and dynamics of folding intermediates become integral to the decision-making process of protein fate. This framework also raises intriguing questions about the interplay between folding kinetics, kinase activity, and degradation pathways. For example, could specific kinases act as "sensors" of misfolded states by targeting cryptic phosphosites exposed during aberrant folding events? Additionally, might cryptic phosphorylation serve as a degradation signal and a means to reroute folding intermediates back toward productive folding pathways under certain conditions? Cryptic phosphorylation as a kinetic checkpoint provides a compelling model for how cells might exploit transient

structural features of folding intermediates to maintain proteostasis.

Our findings also reveal an intriguing link between cryptic phosphosites and cancer. By cross-referencing our dataset of cryptic phosphosites with somatic mutations cataloged in the COSMIC and PTMVar, we identified hundreds of mutations in cryptic phosphosites annotated as disease-related. Notably, some of these mutations are phosphomimetic substitutions, in which amino acids with similar charge characteristics to phosphorylated residues replace cryptic phosphosites. Such a substitution pattern in tumor-suppressor genes directly suggests a possible adaptive advantage in certain cancers, where cryptic phosphosite mutations may reduce protein expression or function, thus enhancing tumor fitness.

The possibility of using advanced path sampling algorithms to accurately predict protein folding mechanisms with an atomic level of resolution and identify the on-pathway folding intermediates has already been exploited to develop a new approach for rational drug discovery termed pharmacological protein inactivation by folding intermediate targeting (PPI-FIT) (Spagnolli et al, 2021). This scheme is based on identifying small molecules that selectively bind to pockets that are transiently solvent-exposed only in on-pathway folding intermediates but become buried in the native state, in close analogy with cryptic phosphosites. The interaction of a small ligand with a folding intermediate prevents the completion of the folding process, thus triggering the protein quality control machinery to send the polypeptide to degradation. The identification of cryptic phosphosites provides an additional perspective to this therapeutic paradigm. Cryptic phosphorylation could directly affect our understanding of diseases associated with protein misfolding and aggregation, offering novel therapeutic angles for conditions such as neurodegenerative disorders, where the balance between proper folding and degradation is altered. Many cancers exhibit dysregulated protein expression and stability, and targeting cryptic phosphosites could be a promising strategy to selectively degrade oncogenic proteins or proteins involved in cancer cell survival pathways. For example, selectively enhancing the phosphorylation of a cryptic site on an oncogenic protein could destabilize it and promote its degradation via the proteasome. Conversely, preventing phosphorylation at a cryptic site on a tumor suppressor (e.g., by inhibiting the specific kinase) could enhance protein stability and restore function.

In summary, our study establishes cryptic phosphosites as critical regulators of protein expression and highlights their potential as therapeutic targets across a wide spectrum of human diseases.

# Methods

### Reagents and tools table

| Reagent/resource | Reference or source | Identifier or catalog number |
| --- | --- | --- |
| **Experimental models** | | |
| HEK-293 cell (*H. sapiens*) | ATCC | CRL-1573 |
| Flp-In™ T-REx™ 293 Cell Line (*H. sapiens*) | Thermo-Fisher | R78007 |
| **Recombinant DNA** | | |

| Reagent/resource | Reference or source | Identifier or catalog number |
| --- | --- | --- |
| pRP[TetOn]-TRE CHK1-FLAG | This study | VB210203-1181cbx |
| pRP[TetOn]-TRE CHK1_T382A-FLAG | This study | VB210203-1183qkc |
| pRP[TetOn]-TRE CHK1_T382D-FLAG | This study | VB210203 1184naa |
| pOG44 Flp-Recombinase Expression Vector | Thermo-Fisher | V600520 |
| pcDNA5/FRT/TO SMAD2-FLAG | This study | VB230303-1038vsk |
| pcDNA5/FRT/TO SMAD2_S417A-FLAG | This study | VB230303-1044atu |
| pcDNA5/FRT/TO SMAD2_S417E-FLAG | This study | VB230303-1046xhp |
| pcDNA5/FRT/TO PYST1-FLAG | This study | VB230308-1261xzm |
| pcDNA5/FRT/TO PYST1_S300A-FLAG | This study | VB230308-1262bdg |
| pcDNA5/FRT/TO PYST1_S300E-FLAG | This study | VB230308-1263zfz |
| **Antibodies** | | |
| Anti-FLAG M2 | Sigma-Aldrich | F1804 |
| goat polyclonal HRP anti-mouse | Jackson ImmunoResearch | 115-035-003 |
| **Chemicals, enzymes, and other reagents** | | |
| Fetal bovine serum | Biowest | S181T |
| Pen/Strep | Gibco | 15140122 |
| Non-essential-aminoacids | Euroclone | ECB3054D |
| L-Glutamine | Gibco | 25030081 |
| DMEM | Gibco | 1195092 |
| Puromycin | Thermo-Fisher Scientific | A1113802 |
| Hygromycin | Invitrogen | 10687010 |
| Doxycicline | Sigma-Aldrich | D9891 |
| Zeocin® (solution) | InvivoGen | ant-zn-1 |
| Blasticidin (solution) | InvivoGen | ant-bl-1 |
| Trypsin | Sigma-Aldrich | T4049-100ML |
| Trans-Blot Turbo RTA Mini 0.2 µm PVDF Transfer Kit, for 40 blots | BIORAD | 1704272 |
| Amersham ECL™ Prime Western Blotting Detection Reagent | GE Healthcare | RPN2236 |
| Amersham ECL™ Select Western Blotting Detection Reagent | GE Healthcare | GERPN2235 |
| Any kD™ Criterion™ TGX Stain-Free™ Protein Gel | BIORAD | #5678124/ #5671125 |
| Cycloheximide | Sigma-Aldrich | 239763-M |
| Lipofectamine™ 2000 | Life Technologies | 11668500 |
| Tween ® 20 | Thermo Scientific | J20605-AP |
| Nonfat dried milk powder | PanReac AppliChem | A0830,0500 |

| Reagent/resource | Reference or source | Identifier or catalog number |
|---|---|---|
| NP40 Tergitol ® solution | Sigma | NP40S-100ML |
| TritonTM X-100 | Sigma | T9284-100ML |
| Dithiothreitol | Sigma-Aldrich | CAS No. 3483-12-3 |
| 4X Laemmli sample buffer | BIORAD | 1610747 |
| Pierce™ BCA Protein Assay Kit | Thermo Scientific | 23225 |
| DMSO | PanReac AppliChem | A3672,0250 |
| **Instruments** | | |
| ChemiDoc Touch Imaging System | BIORAD | |
| Trans-Blot Turbo Transfer System | BIORAD | |
| Criterion™ Vertical Electrophoresis Cell | BIORAD | |
| Varioskan LUX | ThermoFisher Scientific | |
| Software | Source | |
| Snapgene | https://www.snapgene.com/support/downloads | |
| GraphPad PRISM 8 | https://www.graphpad.com/ | |
| SPECTRUS v.Mar2024 | http://spectrus.sissa.it/download.html | |
| python v3.13 | https://www.python.org/downloads/ | |
| mdtraj v1.9 | https://www.mdtraj.org | |
| Gromacs v2018.6, v2024 | https://www.gromacs.org/ | |
| Orthofinder v2.5.5 | https://github.com/davidemms/OrthoFinder | |
| VMD 1.9 | https://www.ks.uiuc.edu/Development/Download/download.cgi | |
| **Other** | | |
| Alphafold2 | https://alphafold.ebi.ac.uk/download | |
| Phosphositeplus | https://www.phosphosite.org/staticDownloads | |
| COSMIC | https://cancer.sanger.ac.uk/ | |
| Uniprot Proteomes | https://www.uniprot.org/proteomes | |

## Assembly of the cryptic phosphosites library

To analyze PTM sites within the human proteome, we extracted structural data from the AlphaFold Protein Structure Database in the form of PDB files which directly provide the atomic coordinates necessary for solvent accessibility estimation. Calculations of solvent-accessible surface area (SASA) for each amino acid of the human proteome were performed using the Shrake-Rupley algorithm integrated within the MDTraj Python library (Bartolucci, 2018). Relative SASA (RSA), a structure-based metric, was calculated using the Shrake-Rupley algorithm by dividing the SASA of a particular residue by the maximum SASA that the same

residue possesses when inserted into a GXG tripeptide (Tien, 2013). RSA values, expected to range between 0 and 1, were considered as a surrogate index of solvent accessibility for each individual PTM residue. PTM data, including methylation, phosphorylation, acetylation, sumoylation, ubiquitination, and glycosylation, were compiled from the PhosphoSitePlus database (Hornbeck, 2015) and cross-referenced using the protein accession IDs to match RSA values with AlphaFold's per-residue model confidence scores (pLDDT) (Jumper, 2021; Varadi, 2024). To ensure the accuracy of our analysis, residues with pLDDT scores below 65, reflecting a low-confidence prediction, were excluded. High-confidence PTM sites were then visualized and sorted using the Python's Seaborn library (Waskom, 2021).

## Molecular dynamics simulations

Molecular Dynamics (MD) simulations were used to investigate the changes in phosphosites solvent exposure over 1 µs. While this time window is small when compared to the inverse rate of many protein conformational reactions, it is sufficiently long to explore possible local rearrangements leading to an increase in solvent exposure. Simulations were conducted with Gromacs 2018.6 and Gromacs 2024.software, using the Charmm36m force field (Huang, 2017) in explicit TIP3P water. We employed the LINCS algorithm (Hess, 1997) to enforce bond length constraints. During the initial equilibration phase, we applied position restraints to heavy atoms to prevent significant structural alterations. We enforced charge neutrality by adding the appropriate number of counterions to reach a final concentration of 150 mM NaCl. We performed energy minimization via the steepest descent method, followed by equilibration under constant volume and temperature (NVT) for 200 picoseconds (ps) at 310 K, and then constant pressure and temperature (NPT) equilibration for another 200 ps. In both equilibrations, we used the V-rescale thermostat (Bussi, 2007) at 310 K and the Parrinello-Rahman barostat at 1 bar (Parrinello, 1981). We generated a single 1 µs-long MD trajectory in NPT conditions at 310 K and 1 bar, with an integration time step of 2 fs. Data were collected every 5 ps. For electrostatic interactions, we used a 12 Å cut-off with Particle mesh Ewald for long-range interactions, while for Van der Waals forces, we resorted to a force-switch modifier between 10 and 12 Å. The analysis of the MD trajectories involved frame-by-frame calculations of the root mean square deviation (RMSD) and the RSA, utilizing Gromacs and the MDTraj Python package, respectively, with the energy-minimized PDB file serving as a reference. Data visualization was performed using the Matplotlib library in Python, and all protein images were produced with Visual Molecular Dynamics (VMD, University of at Urbana Illinois).

## Protein folding simulations

In principle, MD may also be applied to simulate thermally activated structural changes of proteins, including allosteric transitions, protein-ligand binding processes, and protein folding/misfolding events. In practice, however, the applicability of plain MD simulations to protein folding is hindered by the fact that biopolymers are frustrated systems and that completing the protein folding reaction involves overcoming entropy-driven free-energy barriers of many units of thermal energy, $k_BT$. As a consequence,

even generating a single productive folding trajectory would typically require integrating the equations of motion for $o(10^6)$ particles for $10^{12}$–$10^{15}$ time steps. Unfortunately, these numbers are several orders of magnitude larger than those accessible even to the most powerful special-purpose supercomputer (Dingfelder, 2021). To overcome these limitations, we resorted to an enhanced sampling technique based on combining a variational principle, the so-called Bias Functional (BF) method (Ianeselli, 2018), with a special type of biased dynamics called ratchet-and-pawl MD (rMD) (Paci, 1999). rMD is designed to enhance the generation of productive atomistic trajectories connecting from two well-defined conformational states (hereby referred to as the reactant and product, respectively). In a rMD simulation, the system evolves according to plain MD whenever it spontaneously progresses towards the product state, according to a predefined proxy of the reaction coordinate. Conversely, a small history-dependent biasing force switches on whenever the system tries to back track toward the reactant (A Beccara, 2012). A commonly adopted expression for the bias acting on the $i$th atom in the macromolecule is:

$$\mathbf{F}_i^B(X,t) \begin{cases} -k\nabla Q(X)(Q(X) - Q_m(t)), & \text{if} \quad Q(X) \leq Q_m(t) \\ 0 & \text{if} \quad Q(X) \leq Q_m(t) \end{cases} \quad (1)$$

In this equation, $k$ is an arbitrary constant setting the strength of the force, $X(t)$ is the instantaneous molecular configuration (i.e., the set of all Cartesian coordinates of all atoms at time $t$), $Q(X)$ is the proxy of the reaction coordinate (here assumed to depend solely on the molecular conformation X), and $Q_m(t)$ is maximum value attained by Q up to time $t$.

The history-dependent force in Eq. (1) drives the system out of thermal equilibrium. Despite this, it has been shown that, if the bias in Eq. (1) is applied along the *ideal* reaction coordinate $Q(X)$ (namely, the so-called committor function), then the rMD trajectories sample the Boltzmann distribution restricted to the transition region in between the reactant and product states (Bartolucci, 2018).

In practice, since rMD resorts to a proxy of the reaction coordinate, its results are inherently approximate and in particular only provide lower bounds to the free-energy barriers.

Several schemes have been introduced to reduce to minimum systematic errors arising from a suboptimal choice of Q. In particular, the BF approach is based on scoring *a posteriori* each generated rMD trajectory, using a function that estimates the amount of systematic error introduced by the biasing force in each trajectory X(t) :

$$T = \sum_i^N \frac{1}{m_i \gamma_i} \int_0^t d\tau \left| F_i^B[X(\tau), \tau] \right|^2 \quad (2)$$

In this equation, $m_i$ and $\gamma_i$ are the mass and friction constants of the $i$-th atom, while the integral is evaluated along a trajectory $X(\tau)$ generated by rMD, lasting for a time $t$. It was formally shown that trajectories with smaller values of T are more realistic, namely, have a larger probability to occur in the absence of the biasing force. In many applications of the BF scheme, including the present one, the scoring function T is used to identify and discard rMD trajectories that have been particularly heavily biased.

The combination of rMD and BF correction schemes have been successfully applied to the folding of many proteins (Wang, 2016;

Ianeselli, 2018; Wang, 2018; Dingfelder, 2021), using the overlap between the instantaneous and native contact map provides as a proxy of the reaction coordinate (Camilloni, 2011), i.e.,:

$$Q(X) = \sum_{|i-j|>35} \left[ \left( C_{ij}(X^{\text{native}})^2 - \left( C_{ij}(X) - C_{ij}(X^{\text{native}})^2 \right) \right] \quad (3)$$

Here, $C_{ij}(X)$ denotes an entry of the continuous atomic contact map of the protein configuration X. The native contact map is obtained after energy minimizing the experimental crystal or NMR structure retrieved from the protein data bank. The entries $C_{ij}(X)$ are defined to interpolate smoothly between 0 and 1 according to the following switching function:

$$C_{ij}(X) = \begin{cases} \dfrac{1-\left(\frac{r_{ij}}{r_0}\right)^6}{1-\left(\frac{r_{ij}}{r_0}\right)^{10}} & \text{if} \quad r_{ij} < r_c \\ \dfrac{6}{10} & \text{if} \quad r_{ij} < r_0 \\ 0 & \text{if} \quad r_{ij} < r_c \end{cases} \quad (4)$$

In this equation, $r_{ij}$ is the Euclidean distance between the $i$th and the $j$th atom, $r_0$ is a typical distance defying residue contacts (set to 7.5 Å) and $r_c$ is a cut-off distance (set to 12.3 Å) beyond which the contact is smoothly set to 0. Note that the summation in Eq. (3) is restricted to $|i-j| > 35$ to exclude atomic pairs whose motion is highly correlated because of topological proximity. The coupling constant k in Eq. (1), which sets the strength of the biasing force, is chosen to be of $1.0 \times 10^{-4}$ kJ/mol. This set of values ensures that the modulus of the biasing force $F_B = |\sum_i F_i^B(X)|$ is at least one order of magnitude smaller than the modulus of the sum of all physical forces, $|\sum_i F_i(X)|$.

Several MD trajectories were initiated from each unfolded protein conformation, generated by thermal unfolding starting from the native structure. Typically, this was done running 3–5 ns of MD at 800 K, in the NVT ensemble, starting from the energy-minimized native structure.

## Dynamical filtering of post-translationally exposed sites entries

Starting from the database of PTMs of the human proteome, we applied a preliminary set of filters in order to: (i) select only phosphorylation sites; (ii) select putative cryptic phosphosites, which are defined as those having a RSA ≤ 0.15; (iii) select only phosphosites in structured regions, for which AlphaFold predicts a structure with high confidence (per-residue model confidence score ≥65). In principle, the resulting dataset may still contain several post-translationally exposed sites. Indeed, phosphosites that appear to be buried in the static PDB structure may become solvent-exposed during thermal oscillations. To exclude these post-translationally exposed sites from the initial dataset of cryptic phosphosites, we performed a dynamical analysis based on the SPECTRUS package (McGibbon, 2015). This software exploits a specific elastic network model to efficiently characterize structural fluctuations in the native state via normal-mode analysis (Hunter, 2007; Tirion, 1996; Micheletti, 2013). By comparing the relative weights of the normal modes, SPECTRUS infers alternative decompositions of the protein structure in quasi-rigid sub-domains (Potestio et al, 2009). These alternative

decompositions are then ranked according to a quality score that pinpoints significant subdivisions based on the balance of intra- and inter-domain distance fluctuations compared to a random reference case (Ponzoni et al, 2015). Since normal-mode analysis is ill-defined in unstructured regions, we preliminarily removed low-confidence segments from our initial protein dataset, typically loops, linkers, or short disorder regions, having an AlphaFold score lower than 65. The resulting dataset included individual protein domains or contiguous domains considered as a single entry if they had at least five residues within 5 Å from each other. Cryptic phosphosites lying within protein domains encompassing less than 40 residues were excluded from the analysis, due to the low reliability of normal-mode analysis to characterize the internal dynamics of short polypeptides. The RSA value of cryptic phosphosites was then recalculated on the new set of entries, filtering out residues showing a new RSA value higher than 0.15. Finally, SPECTRUS identified quasi-rigid domains among the new protein dataset by generating β-Gaussian models for each entry and performing a spectral decomposition based on such model (Micheletti et al, 2004). Cryptic phosphosites lying within quasi-rigid domains belonging to the subdivision with the highest SPECTRUS relative quality score were eliminated when their side chain made less than 80% of intra-domain contacts. The proportion of intra-domain contacts (Pidc) was computed as the proportion of residues within 6 Å from the side chain of a phosphosite that belong to the same quasi-rigid domain of the phosphosite.

## Plasmids

All expression plasmids were purchased from VectorBuilder. For human SMAD2 (UniProt ID: Q15796), three distinct versions of the gene were designed and codon optimized using the Vector-Builder online tool: the wild-type (WT) version (VB230303-1038vsk), the phosphoablative mutant (S417A) (VB230303-1044atu), and the phosphomimetic mutant (S417E) (VB230303-1046xhp). Three versions were also designed for human PYST-1 (UniProt ID: Q16828): the WT version (VB230308-1261xzm), the phosphoablative mutant S300A (VB230308-1262bdg), and the phosphomimetic mutant S300E (VB230308-1263zfz). Each gene variant was fused to a FLAG tag sequence (DYKDDDDK) for antibody recognition and discrimination from the endogenous counterpart and inserted in a pcDNA5/FRT/TO expression vector, designed for use with the Flp-In T-REx expression system. The three versions of the CHK1 protein (UniProt ID: O14757) included the WT sequence (VB210203-1181cbx), the phosphoablative mutant T382A (VB210203-1183qkc), and the phosphomimetic mutant T382D (VB210203 1184naa). Each gene variant was fused to a FLAG tag sequence (DYKDDDDK) at the C-terminus. All the engineered proteins were expressed under the control of a Tet-inducible promoter (pRP[TetOn]-TRE).

## Cells and treatments

Cells were grown in Dulbecco's Minimal Essential Medium (DMEM Gibco, #11965092), supplemented with 10% heat-inactivated fetal bovine serum (FBS) Tetracycline Free (Biowest, #S181T), Penicillin/Streptomycin (Pen/Strep, Gibco, #15140122), non-essential amino acids (NEAA, Euroclone, #ECB3054D), and L-Glutamine (Gibco, #25030081). Cells were maintained in 100 mm Petri dishes, with routine subculturing every 3–4 days.

To ensure consistency in experimental conditions, the cells used in the experiments were not passed more than 15 times. HEK293 cell line was obtained from ATCC (ATCC CRL-1573). Flp-In T-REx 293 cells, obtained from ThermoFisher Scientific (R78007), were cultured with medium containing 100 µg/mL zeocin (InvivoGen, ant-zn-1) and 10 µg/mL blasticidin (InvivoGen, ant-bl-1) for selection and maintenance. All plasmids were transfected using Lipofectamine 2000 (Life Technologies), following the manufacturer's instructions. Forty-eight hours after transfection, cells were selected with the appropriate selection medium (200 µg/mL hygromycin for CHK1 constructs, 100 µg/mL hygromycin, and 10 µg/mL blasticidin for PYST-1 and SMAD2) until stable pools were obtained. Cells were then counted and seeded in 100 mm Petri dishes at a very low confluence ( ~ 100–200 cells per dish) and maintained in a selection medium, which was changed every 2–3 days, until cellular clones were clearly visible by eye. By using a pipette tip each clone was scraped from the dish and moved to a 48-well plate first, and a six-well later, for recovery and expansion. Once clones were stabilized, they were frozen and stored at −80 °C or liquid nitrogen tanks. Doxycycline, acquired in powdered form (Sigma-Aldrich, D9891), was dissolved in sterile-filtered water to create a 100 mg/mL stock solution. Dilutions were made to obtain single-use aliquots of 1 mg/mL. All solutions were kept at −20 °C. Further dilutions produced in single-use aliquots of 1 mg/mL were stored at −20 °C. Treatment with doxycycline entailed diluting the stock solution in the medium before applying it to the cells, ensuring accurate dosage. Cycloheximide, obtained in powdered form (Sigma-Aldrich, 239763-M), was dissolved in DMSO to a concentration of 50 mg/mL and subsequently aliquoted for storage. These aliquots were preserved at −20 °C. The working concentration employed in experiments was 50 µg/mL, achieved by a 1:1000 dilution, which also maintained a 0.1% DMSO concentration (w/v) in the medium.

## Western blotting

Cellular samples were lysed using a buffer composed of 10 mM Tris (pH 7.4), 0.5% NP-40, 0.5% Triton X-100, 150 mM NaCl, supplemented with cOmplete™ Protease Inhibitor Cocktail Tablets (Roche, #11697498001). Following lysis, lysates were centrifuged at $20,000 \times g$ for 1 min, after which the supernatants were harvested. The protein concentration within these supernatants was then determined utilizing the BCA protein assay kit (ThermoFisher Scientific, #23225). For subsequent analyses, lysates were prepared by diluting samples to contain 20–30 µg of total protein in a 2:1 ratio with 4× Laemmli sample buffer (Bio-Rad, #1610747) enriched with 100 mM Dithiothreitol (DTT, CAS No. 3483-12-3, Sigma-Aldrich). Samples were then heated at 95 °C for 8 min to ensure denaturation. Proteins were resolved according to their molecular weights under denaturing conditions using sodium dodecyl sulfate-polyacrylamide gel electrophoresis (SDS-PAGE). All the samples were loaded into pre-cast gels (Bio-Rad, Any kD™ Criterion™ TGX Stain-Free™ Protein Gels, #5678124/#5671125) and subsequently electro-transferred onto polyvinylidene difluoride (PVDF) membranes employing the Trans-Blot Turbo RTA Mini 0.45 µm LF Transfer Kit (Bio-Rad, #1704274). To prevent non-specific binding, membranes were incubated for 30–60 min in a blocking solution consisting of 5% (w/v) nonfat dry milk dissolved in Tris-buffered saline with 0.01% Tween (TBST). Blots were incubated with the

primary antibody (anti-FLAG M2, code F1804, Sigma-Aldrich, diluted 1:1000 in blocking solution) overnight at 4 °C. Membranes were then washed three times with TBST, each lasting 10 min. Subsequently, the membranes were incubated with a horseradish peroxidase-conjugated secondary antibody (goat polyclonal HRP Anti-mouse, code 115-035-003, Jackson ImmunoResearch, diluted 1:10,000 in blocking solution) for 1 h at room temperature. After three additional washes with TBST and a final rinse with Milli-Q water, detection of the protein bands was achieved using either the Amersham ECL Prime Western Blotting Detection Kit (GE Healthcare, #RPN2232) or the Amersham™ ECL Select™ Western Blotting Detection Reagent (#RPN2235), depending on the required signal intensity. Visualization of the signals was performed using the ChemiDoc Touch Imaging System (Bio-Rad). The final quantification of proteins was achieved through densitometric analysis of the corresponding western blot bands. Each signal was normalized against the corresponding total protein lane, determined utilizing the enhanced tryptophan fluorescence technology of stain-free gels (Bio-Rad). For clonal analysis, each signal was normalized on the total protein loaded in the lane, further divided by the loading reference to control image exposure, allowing the cross-comparison of different gels. Only for CHK1, signals were also normalized on the corresponding non-induced samples to control expression leakage. The expression threshold to discriminate positive and negative clones was set at the first quartile, that is the value below which 25% of all the data points for each protein fall.

## Phylogenetic analyses

To perform a comparative genomics study of cryptic phosphosites, we utilized OrthoFinder2, a robust platform for comparative genomic analysis (Emms and Kelly, 2019). We included proteomes from 137 eukaryotic species downloaded from the UniProt database, selected to closely align with the phylogenetic lineage of Homo sapiens, such as members of the Primates order, to ensure relevance in our orthologs analysis. The selection criteria were as follows:

- Eukaryotic Focus: Only eukaryotic organisms were chosen due to the complex network of PTMs being absent in Bacteria and Archaea.
- Proteome Status: Each organism's proteome had to be designated as either a "Representative proteome" or a "Reference proteome" the latter of which represents a comprehensive cross-section of the taxonomic diversity within UniProtKB.
- Proteome Completeness: We used the Benchmarking Universal Single-Copy Orthologs (BUSCO) score as a measure of proteome completeness. We calculated the Interquartile Range (IQR) from the 25th to the 75th percentile of selected species' BUSCO scores and excluded any organism whose BUSCO score fell below the lower limit, which was defined as 1.5 times the IQR below the 25th percentile. Starting from the OrthoFinder results previously generated, the protein sequences of each human orthogroup were aligned using Clustal Omega, a widely used bioinformatics tool utilized for performing multiple sequence alignment (MSA). These analyses allowed us to assign an entropic score (ES) value to each PTM. The ES quantifies the level of disorder or randomness within a system. In relation to protein structures, it measures the variability in the aligned positions of amino acid residues. Higher

scores reflect greater disorder, while lower scores suggest increased order and stability, which in turn indicates greater conservation of residues. The entropy definition adopted to quantify the residues' level of evolutionary conservation is:

$$S = \frac{-\sum_{\alpha=1}^{K} p_\alpha \log p_\alpha}{\log(K)} \tag{5}$$

$K$ represents the total number of distinct residue types observed at the aligned position. This includes the 20 standard amino acids, the gap symbol, and any ambiguous residue type present in the selected proteome entries (e.g., B, Z, X). $p_\alpha$ is the relative frequency of residue type α at the aligned position, calculated as the count of residue type α divided by the total number of sequences in the alignment.

According to the mathematical definition of entropic score, the limit case in which, within each orthogroup, the phosphorylated amino acid is replaced by a completely random residue yields an entropic score of 1. The opposite limit, in which all members of the orthogroups have the same amino acid in the position of the phosphorylated amino acid, yields an ES of 0. Indeed, the denominator, ln(K), normalizes the entropy to a scale between 0 and 1, where 0 indicates perfect conservation (a single residue type dominates) and 1 indicates maximum variability (all K residue types are equally represented).

## Cross-integration with the COSMIC dataset

To perform a preliminary exploration of the pathological relevance of mutations occurring in cryptic phosphosites, the COSMIC dataset (https://cancer.sanger.ac.uk/cosmic) and the PTMVar dataset (https://www.phosphosite.org/staticDownloads) were analyzed in order to retrieve mutations occurring in cryptic phosphosites. To this end, the COSMIC targeted screen mutants TSV file (release v100, GRCh38) was preprocessed to eliminate the lines relative to negative screens and to count how many times a given mutation is present in the dataset. This analysis generated a VCF-like file from which the data about proteins containing phosphosites were extracted. Next, we assembled two different tables, one for cryptic and another for non-cryptic phosphosites, each containing the following pieces of information:

- Gene subACCID: principal gene access ID from UniProt (UniProtKB ID).
- Gene symbol: principal gene name from UniProt.
- Position: The number of the mutated residues in the primary sequence.
- Mutation CDS: a change that has occurred in the nucleotide sequence as a result of the mutation.
- Mutation Amino Acid: a change that has occurred in the peptide sequence as a result of the mutation.
- Count: The number of unique mutated samples for this mutation.
- Mutation Type: The type of mutation (substitution, deletion, insertion, complex, fusion etc.).

Afterward, the data about mutations occurring in cryptic phosphosites and annotated as cancer-related were extracted from the bulk table. Phosphomimetic mutations were arbitrarily limited those that transform the amino acid found at the phosphosite (S, T or Y) into an aspartic acid (D) or a glutamic acid (E). The PTMVar dataset was filtered similarly to extract mutations in cryptic

phosphosites and among them phosphomimetic mutations. The tables with the final entries are available in the supporting information. All the analyses were performed using custom Python scripts (provided as online extended material).

## Data availability

All the data that support the findings of this study are available within the manuscript, extended figures, tables, or from the corresponding authors upon request. The main dataset, source and raw data, and all the codes employed in this study are accessible through Zenodo at the following link: https://zenodo.org/records/16904945.

The source data of this paper are collected in the following database record: biostudies:S-SCDT-10_1038-S44318-025-00567-1.

## Peer review information

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

## Acknowledgements

We thank all the members of the Dulbecco Telethon Laboratory of Prions & Amyloids at Department CIBIO, University of Trento (www.cibio.unitn.it/95/dulbecco-telethon-laboratory-of-prions-and-amyloids) for the fruitful discussion of the results presented in this manuscript. This work was supported by grants from Fondazione Telethon (GGP20043, GMR24T2072) to E.B. We acknowledge financial support under the National Recovery and Resilience Plan (NRRP), Mission 4, Component 2, Investment 1.1, Call for tender No. 104 published on 2.2.2022 by the Italian Ministry of University and Research (MUR), funded by the European Union, NextGenerationEU Project Title "Functional Role of Protein folding Intermediates: *A Cross-Disciplinary Study Integrating Molecular Simulations with Biophysical and Biochemical Experiments*", CUP H53D23000860006, Grant Assignment Decree No. 1064 adopted on 18.07.2023 by the Italian Ministry of Ministry of University and Research (MUR) to PF. DG was sponsored by a fellowship from the National Recovery and Resilience Plan (NRRP), Mission 4, Component 2, Investment 3.3, Call for tender published on 14/07/2021 by the Italian Ministry of University and Research (MUR), funded by the European Union and co-funded by Sibylla Biotech Spa. JRR was supported by a grant from the Spanish National Research Agency (AEI), with participation of EU FEDER funds (Grant number: PID2020-117465GB-I00).

## Author contributions

**Dino Gasparotto**: Conceptualization; Data curation; Formal analysis; Supervision; Methodology; Writing—original draft; Writing—review and editing. **Annarita Zanon**: Conceptualization; Data curation; Formal analysis; Investigation; Methodology; Writing—original draft; Writing—review and editing. **Valerio Bonaldo**: Conceptualization; Formal analysis; Supervision; Investigation; Methodology; Writing—review and editing. **Elisa Marchiori**: Conceptualization; Formal analysis; Investigation; Writing—review and editing. **Massimo Casagranda**: Formal analysis; Writing—review and editing. **Erika Di Domenico**: Formal analysis; Investigation. **Laura Copat**: Formal analysis; Investigation. **Tommaso Fortunato Asquini**: Conceptualization; Formal analysis; Investigation; Writing—review and editing. **Marta Rigoli**: Data curation; Supervision. **Sirio Vittorio Feltrin**: Data curation; Investigation. **Nuria Lopez Lorenzo**: Conceptualization; Investigation; Writing—review and editing. **Graziano Lolli**: Conceptualization; Writing—review and editing. **Maria Pennuto**: Conceptualization; Writing—review and editing. **Jesùs R Requena**: Investigation; Writing—review and editing. **Omar Rota Stabelli**: Conceptualization; Supervision; Writing—review and editing. **Giovanni Minervini**: Conceptualization; Supervision; Writing—review and editing. **Cristian Micheletti**: Conceptualization; Resources; Supervision; Writing—review and editing. **Giovanni Spagnolli**: Conceptualization; Supervision; Investigation; Writing—review and editing. **Pietro Faccioli**: Conceptualization; Formal analysis; Supervision; Funding acquisition; Writing—original draft; Project administration; Writing—review and editing. **Emiliano Biasini**: Conceptualization; Data curation; Formal analysis; Supervision; Funding acquisition; Validation; Investigation; Writing—original draft; Project administration; Writing—review and editing.

Source data underlying figure panels in this paper may have individual authorship assigned. Where available, figure panel/source data authorship is listed in the following database record: biostudies:S-SCDT-10_1038-S44318-025-00567-1.

## Disclosure and competing interests statement

The authors declare the following competing interests: GS, GL, PF, and EB are co-founders and shareholders of Sibylla Biotech SRL (www.sibyllabiotech.it). The company exploits the information arising from folding pathway reconstruction for drug discovery in a wide variety of human pathologies.

# Expanded View Figures

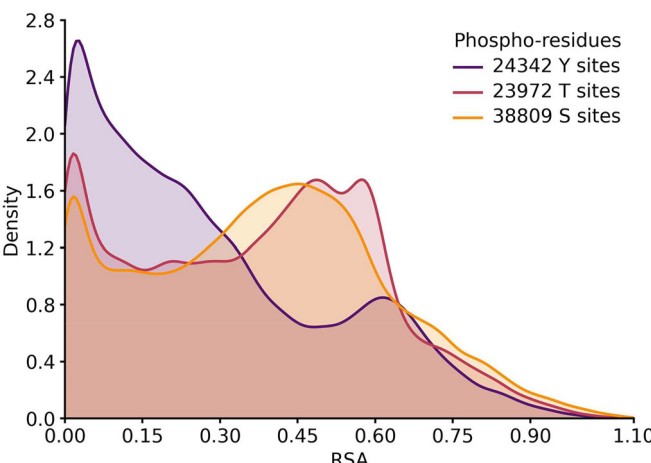

**Figure EV1.**  Graphic distribution of the occurrence of phosphorylation in relation to the RSA of the corresponding target amino acids (serine, tyrosine and threonine).

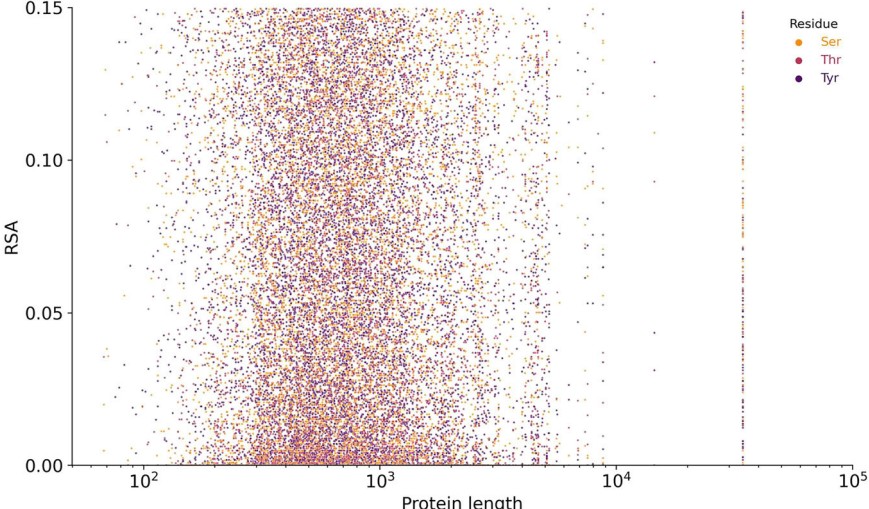

**Figure EV2.** Graphical plot of the RSA of cryptic phosphosites vs the length of the proteins in which they occur.

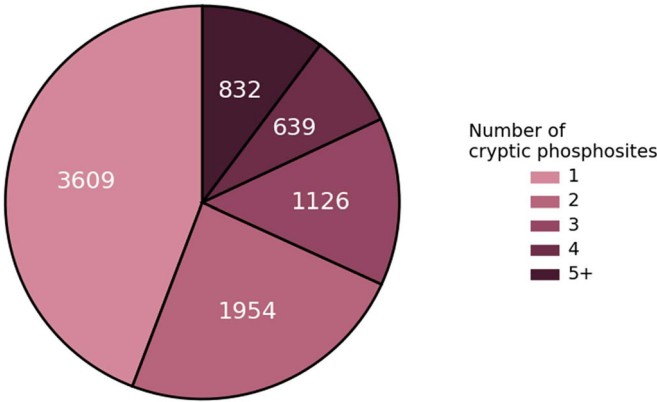

**Figure EV3.** Pie chart illustration of the number of cryptic phosphosites per protein.

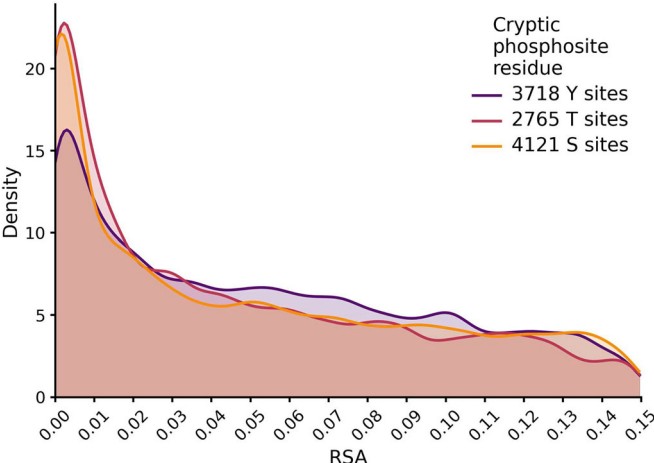

**Figure EV4.** Graphic distribution of the occurrence of phosphorylation in relation to the RSA of the corresponding amino acids after applying the dynamic and structural filtering.

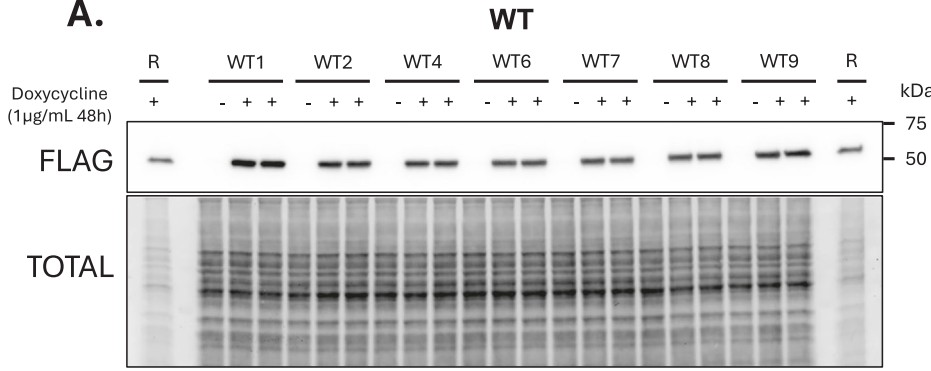

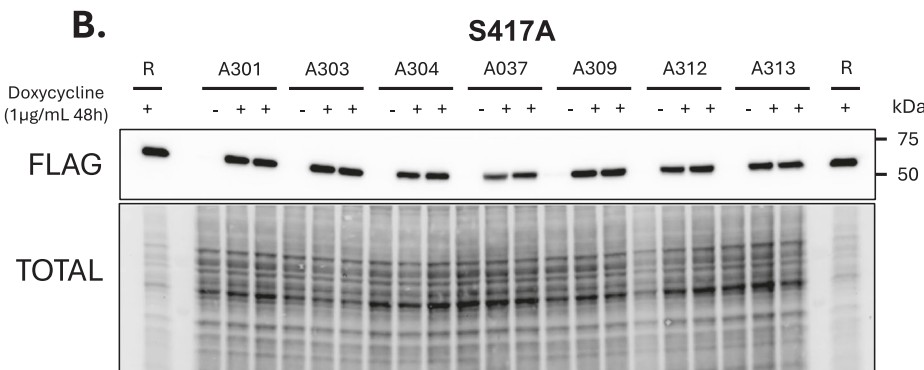

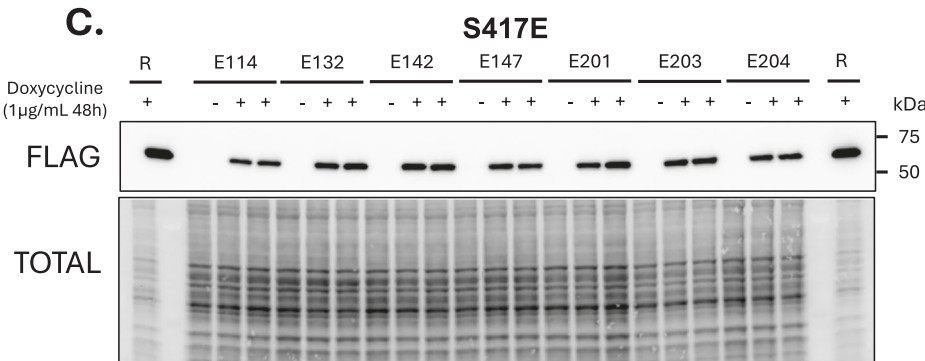

**Figure EV5. Representative western blots of the effect of phosphoablative and phosphomimetic mutations in SMAD2.**

Individual clones (coded with the indicated numbering) stably transfected with WT (**A**), phosphoablative (**B**) and phosphomimetic (**C**) SMAD2 constructs were non-induced (−) or induced with doxycycline for 48 h. A reference clone (R) induced for 48 h was included for comparison among different western blots. Protein expression was detected with an anti-FLAG antibody. Signals were normalized on the corresponding total protein lanes.

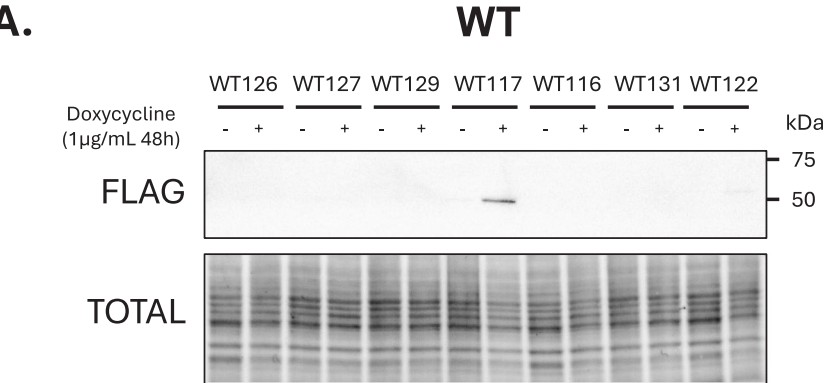

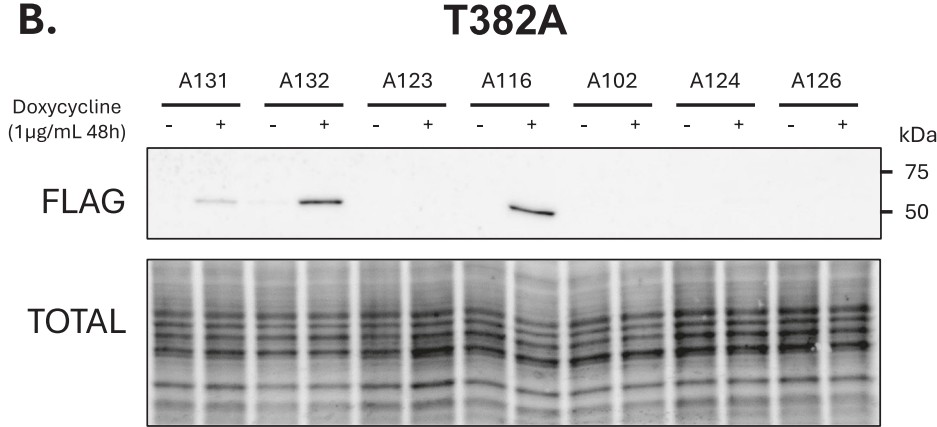

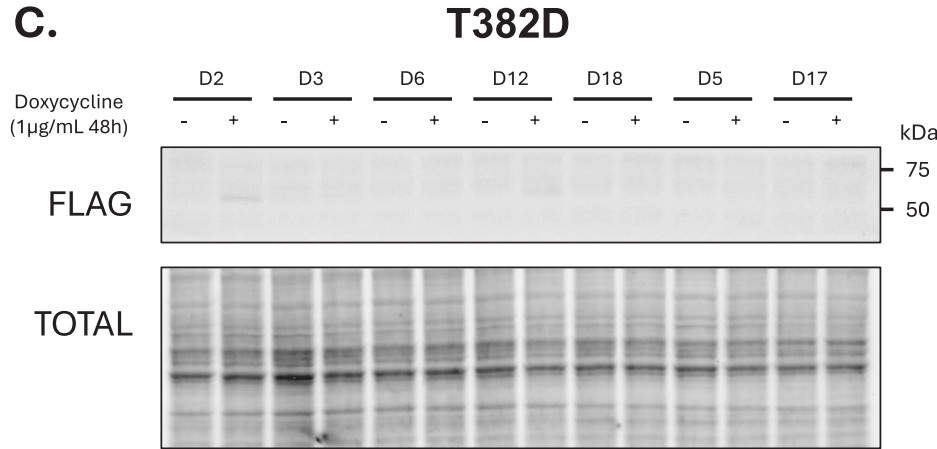

**Figure EV6.  Representative western blots of the effect of phosphoablative and phosphomimetic mutations in CHK1.**

Individual clones (coded with the indicated numbering) stably transfected with WT (**A**), phosphoablative (**B**) and phosphomimetic (**C**) CHK1 constructs were non-induced (—) or induced with doxycycline for 48 h. Protein expression was detected with an anti-FLAG antibody. Signals were normalized on the corresponding total protein lanes.

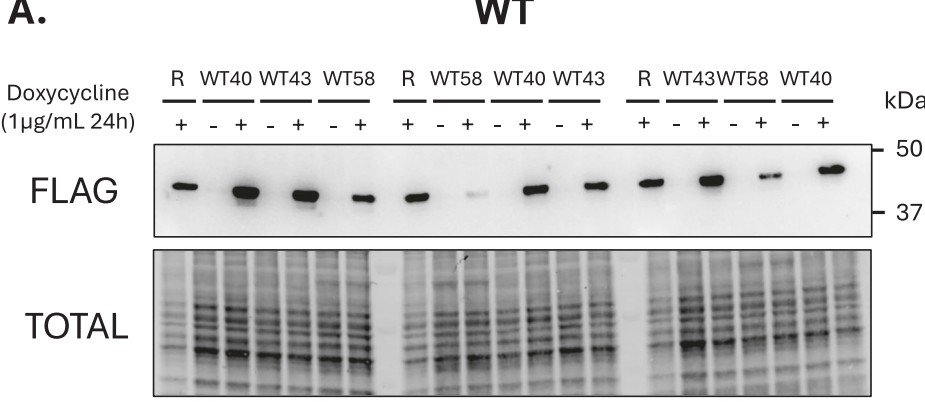

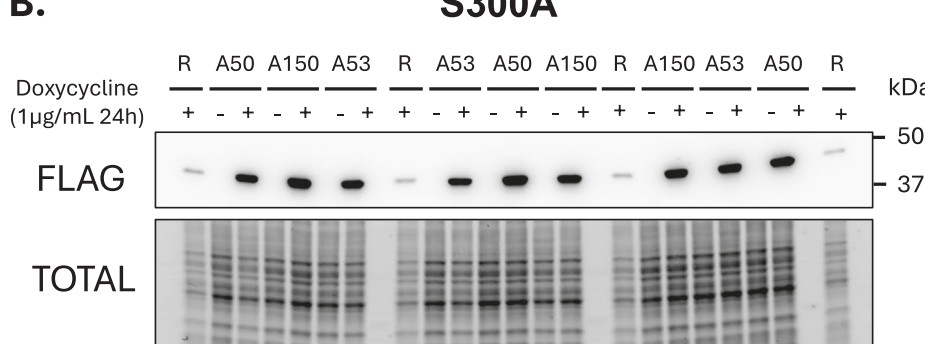

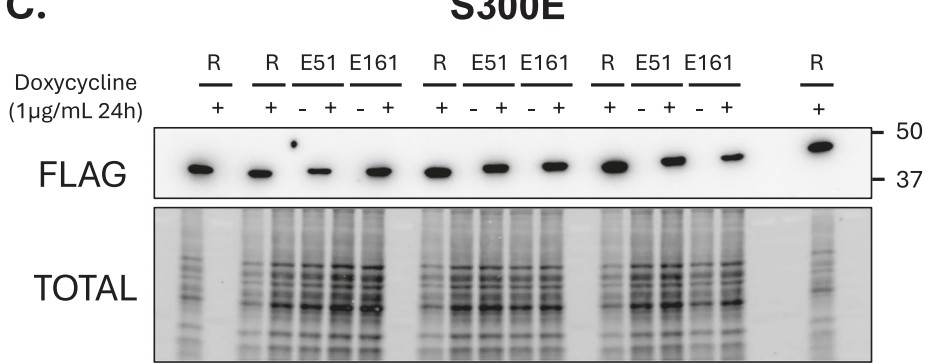

**Figure EV7. Representative western blots of the effect of phosphoablative and phosphomimetic mutations in Pyst1.**

Individual clones (coded with the indicated numbering) stably transfected with WT (**A**), phosphoablative (**B**) and phosphomimetic (**C**) Pyst1 constructs were non-induced (−) or induced with doxycycline for 24 h. A reference clone (R) induced for 24 h was included for comparison among different western blots. Protein expression was detected with an anti-FLAG antibody. Signals were normalized on the corresponding total protein lanes.

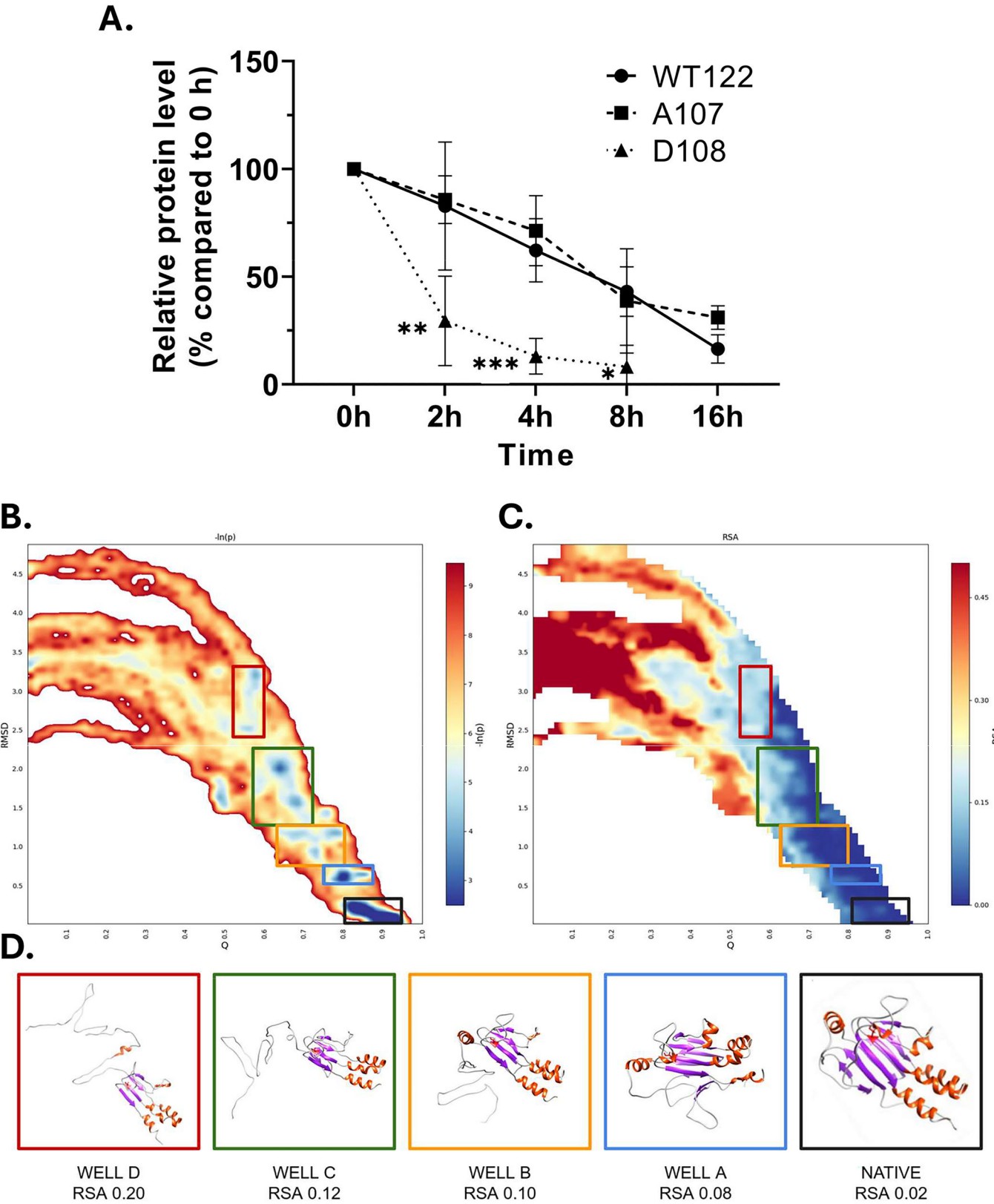

◀ **Figure EV8. Experimental and computational investigation of cryptic phosphosites in CHK1 and SMAD2.**

(A) The analysis of CHK1 stability presented in Fig. 5A–C are plotted together to better represent the statistical differences between the phosphomimetic mutant D382 and the WT T382 and phosphoablative mutant A382. The graph illustrates the quantified data (mean ± standard deviation), normalized to the $t_0$ level. The degradation kinetics were modeled using a one-phase decay curve from which half-time ($t_{1/2}$) values were derived. Data analysis and curve fitting were performed using GraphPad Prism software. Statistical differences vs the WT T382 were obtained by ordinary one-way ANOVA test; *$P < 0.05$; **$P < 0.01$; ***$P < 0.001$. (B) The plot shows the atomistic reconstruction of the SMAD2 folding pathway. Lower-bound approximation of the transition path energy is related to the folding of SMAD2. The energy is plotted as the negative logarithm of the probability distribution ($-\ln(p)$), which is expressed as a function of the collective variables Q and RMSD (65×65 bin matrix). Gaussian blur was applied. The highly populated native state appears as expected in the bottom-right corner (high Q and low RMSD, black rectangle). The indexed squares define the most-populated partially unfolded regions of interest ($-\ln(p)$). (C) The plot shows the distribution of RSA of the S417 residue along the transition path. The highest values are associated with the unfolded state (top left), while in the native state the RSA is consistently below 0.15. (D) Representative conformations for each cluster. S417 is displayed in red, α-helices are colored in orange, β-sheets in light purple.

**A.**

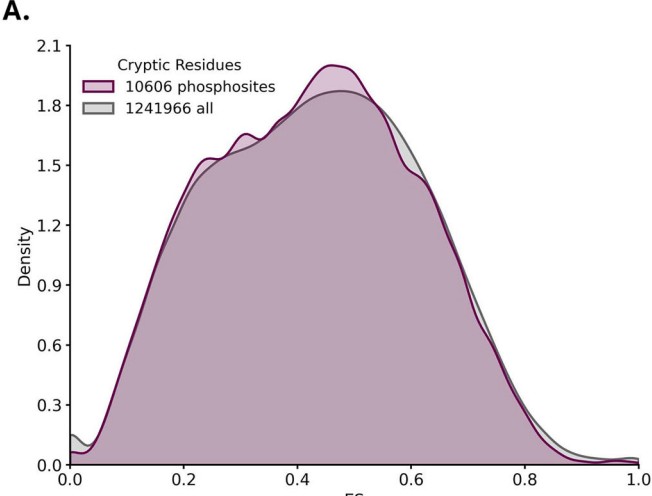

**B.**

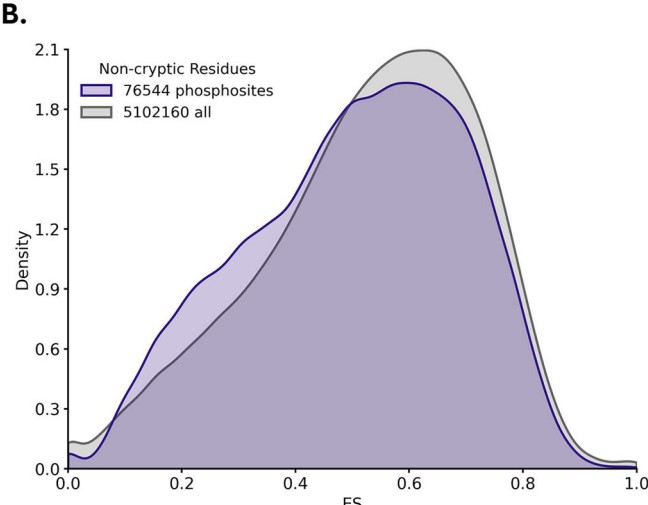

**Figure EV9.  Graphic distribution of evolutionary conservation of cryptic and non-cryptic phosphosites.**

(A-B) KDE plot reports the distribution of ES for cryptic (light purple) and non-cryptic (dark purple) phosphosites. The evolutionary conservation of cryptic and non-cryptic phosphosites were superimposed to the distribution of cryptic and non-cryptic amino acids.

