## [Peer Review File · The EMBO Journal]

Mapping Cryptic Phosphorylation Sites in the Human Proteome

Dino Gasparotto, Annarita Zanon, Valerio Bonaldo, Elisa Marchiori, Massimo Casagrande, Erika Di Domenico, Laura Copat, Tommaso Asquini, Marta Rigoli, Sirio Feltrin, Nuria Lorenzo, Graziano Lolli, Maria Pennuto, Jesus Requena, Omar Stabelli, Giovanni Minervini, Cristian Micheletti, Giovanni Spagnolli, Pietro Faccioli, and Emiliano Biasini

Corresponding author(s): Emiliano Biasini (emiliano.biasini@unitn.it) , Pietro Faccioli (pietro.faccioli@unimib.it)

Review Timeline:

Transfer from Review Commons:	6th Jun 25
Editorial Decision:	11th Jul 25
Revision Received:	25th Jul 25
Accepted:	2nd Sep 25

Review
COMMONS

Editor: Hartmut Vodermaier

Transaction Report: This manuscript was transferred to The EMBO JOURNAL following peer review at Review Commons.

Review #1

1. Evidence, reproducibility and clarity:

Evidence, reproducibility and clarity (Required)

****Summary:****

This preprint uses bioinformatic and experimental approaches to explore the prevalence and consequences of the phosphorylation of residues normally buried in the hydrophobic core of proteins. By cross-referencing validated human phosphosites (PhosphositePlus) with the predicted 3D structures of the human proteome (from the AlphaFold predicted protein structure database), they identified potential "cryptic" phosphosites not expected to be solvent-accessible. They further refined the list using a variety of tools and conclude that a significant percentage (roughly 25%) of known phosphosites in folded domains are cryptic. They go on to experimentally test the consequences of mutating several of these sites in known proteins either to non-phosphorylatable or phospho-mimetic residues, and found that the phosphomimetic mutants had lower half-lives and average expression levels than either the wt or non-phosphorylatable versions. Finally, they show that putative cryptic phosphorylation sites are more highly conserved than those that are surface-accessible, and that some of these cryptic sites are found in tumor suppressor genes and that phosphomimetic mutations at these sites can be found in tumor mutation databases.

****Major comments:****

Overall the experimental approach is relatively straightforward, and in general the authors' interpretation of the results seems reasonable. There were several areas where I believe additional analysis or discussion might clarify the interpretation, however.

1. It would be helpful if the authors could discuss whether there is any correlation between cryptic sites and the extent of experimental validation in the Phosphosite database (e.g. those that were only identified in one or a few MS experiments). It is difficult to determine stoichiometry of phosphorylation experimentally, but can any inference be made on the extent of phosphorylation of cryptic sites vs. more conventional sites located in IDRs or on the surface of globular domains?
2. The authors note that a larger percentage of tyrosine phosphorylation sites are cryptic compared with serine/threonine sites. I assume that tyrosine itself is more highly enriched in the hydrophobic cores of proteins relative to serine or threonine, due to its bulky hydrophobic side chain. Is the increased proportion of cryptic tyrosine phosphorylation

sites more, less, or the same as the proportion of tyrosine in hydrophobic cores relative to serine and threonine?

3. Fig. 5D and E: I had some trouble interpreting these figures. Indicating where the native state is in the plots would be helpful (stated in text as lower right, but a rectangle on the plot would make this more obvious). The text discusses three metastable intermediates, but what is the fourth one shown on the figures (well A, close to the native state)? This could be more explicitly explained.

4. The fact that phosphomimetic mutations of cryptic sites in SMAD2 and CHK1 lead to lower expression levels and shorter half-lives is not surprising, given the expected disruption of the hydrophobic core by introduction of a charged residue. The results certainly show that if phosphorylated, these sites would decrease expression and half-life. With respect to half-life, however, if the authors are correct and cryptic sites are predominately phosphorylated co-translationally, one would expect that the half-life curves for the wt protein would not be a simple exponential, but would instead reflect two distinct populations: those that are phosphorylated during translation, and are almost immediately degraded, and those that escape phosphorylation and have the same half-life as the non-phosphorylatable mutant. Are the actual experimental results consistent with this two-population model? If not, this would be evidence that some of these cryptic sites can be exposed post-translation, either by thermal fluctuation or biological interactions.

5. The authors make a point that cryptic phosphosites are more highly conserved than non-cryptic phosphosites, but it is not clear to me whether it is the side chain itself or its ability to be phosphorylated that is conserved. Supplemental Fig. 9, if I am interpreting it correctly, would suggest it is the residue itself and not its phosphorylation that is conserved. If so, wouldn't this suggest that phosphorylation of these cryptic sites is just an inevitable consequence of the conservation of serine, threonine, and tyrosine residues in hydrophobic core regions? If the authors have evidence that argues against this simple hypothesis, they should discuss it (e.g., cryptic phosphosites are more highly conserved in some cases than non-phosphorylated tyrosine, serine, and threonine residues that are not solvent accessible).

6. Regarding the evolutionary conservation of cryptic sites, have the authors taken into consideration that tyrosine-specific kinases, phosphatases, and reader domains first appeared in the first metazoans, and are for the most part not seen in non-metazoan eukaryotes? I notice some of the proteomes used for the conservation analysis include plants and yeast, which lack most tyrosine phosphorylation.

7. I find the argument that phosphorylation of exposed core residues is part of normal protein quality control/proteostasis to be convincing. Can the authors provide any experimental evidence to support this model (for example, greater phosphorylation of cryptic sites under stress conditions)? I don't think these experiments are necessary, but

would seem to be a logical next step and could be done quite easily through collaboration. 8. The authors note at the end of the discussion that targeting cryptic phosphosites might be a strategy to selectively degrade some proteins in cancer. Practically, how would this work? I can't think of how, but perhaps the authors can provide more specific suggestions.

****Minor comment:****

1. Introduction: "It involves the addition of a phosphate to an hydroxyl group found in the side chain of specific amino acids, typically serine, threonine or tyrosine residues." Of course serine, threonine, and tyrosine are the only standard amino acids with a simple hydroxyl group, so "typically" is not needed here.

2. Significance:

Significance (Required)

In my view this is an important study, bringing rigor and a broad proteomic perspective to a phenomenon that (to my knowledge) had not been carefully examined previously. In terms of the big picture, I am of two minds. On the one hand, showing that phosphorylation of hydrophobic core residues exposed during translation or the early stages of folding can regulate steady state levels of some proteins provides an intriguing new mechanism to control the complement of proteins in the cell, and is potentially an area of regulation in normal physiology or in disease. On the other hand, if this is just part of the normal proteostatic mechanisms (hydrophobic core residues exposed for too long consign the protein to degradation, before it can lead to aggregation and other problems), that is a little less interesting to me. I think future work to tease out whether this mechanism is actually regulated and used by the cell to transmit information will be key. But the first step is showing that the phenomenon is real and widespread, and in my view this preprint accomplishes that goal very well.

I come from a background of studying post-translational modifications in signaling, hence my hope that a regulatory role can be found. But even if cryptic phosphorylation turns out to be unregulated, the work provides important new insight into normal proteostasis, and therefore is a valuable contribution. I should note that I don't have extensive expertise in bioinformatic methods or the computational tools to study protein dynamics, but I assume other reviewers will critically evaluate these methods.

3. How much time do you estimate the authors will need to complete the suggested revisions:

Estimated time to Complete Revisions (Required)

(Decision Recommendation)

Less than 1 month

4. Review Commons values the work of reviewers and encourages them to get credit for their work. Select 'Yes' below to register your reviewing activity at Web of Science Reviewer Recognition Service (formerly Publons); note that the content of your review will not be visible on Web of Science.

Yes

Review #2

1. Evidence, reproducibility and clarity:

Evidence, reproducibility and clarity (Required)

Review on Gasparotto et al "Mapping Cryptic Phosphorylation Sites in the Human Proteome"

Gasparotto et al assess the solvent accessibility of 87,138 post-translationally modified amino acids in the human proteome (from phosphosite plus). Their initial observation is that a large fraction of modified sites are buried, a finding that is pronounced for phosphorylation but not other modifications. Their approach is using alpha fold 3D structures (0.65 cut off) and RSA prediction to get a set of buried sites. Further refinement includes the removing of low-confidence segments (such as loops, linkers, or short disordered regions) and to use SPECTRUS to identify quasi-rigid domains. The idea is that quasi rigid domains may not breathe and thus will be modified during the synthesis or folding.

They generated a final dataset of 10,606 cryptic T, S and Y phospho-sites in 5,496 proteins and state that: "These data indicate that ~5% of all known phospho-sites are cryptic. Impressively, the number translates to ~33% of phosphorylated proteins in the human proteome presenting at least one cryptic phospho-site."

They focus on S417 of the SMAD2, T382 of Chk1, known to be associated with loss of

function effects or proteasomal degradation and S300 of PYST1 negative control. They stably express these proteins as phospho-mimicry or alanine substitution in HEK293. Expression levels were reduced in the phospho-D- mutant versions and upon cycloheximide treatment a reduction of the turnover time for the phospho-D CHK1 was observed. I think we are looking a large clonal difference in the supplemental figures.

The examples are supported by MD simulations that suggest that cryptic phospho-sites can occur during the folding process and affect protein homeostasis by drastically increasing degradation rate and leading to rapid turnover; Essentially the phospho-versions show a solvent exposure.

Evolutionary comparison whether cryptic and non-cryptic sites are differently conserved. Two distinct distributions for cryptic and non-cryptic phospho-sites are observed and Figure 6 shows two entropy distributions of cryptic v non-cryptic. Here it is unclear whether this is significant given the different distributions of the two types when non modified. Finally, overlay of the sites with cancer mutations lists 221 mutations in COSMIC associated with cryptic phosphosites that have been annotated as cancer-related and 138 mutations in PTMVar linked to cancer and other human pathologies.

The identification of buried modification sites and what the biological meaning / implications are is a very interesting topic. However PTM distribution on proteins is very skewed (many papers have identified cluster, hot spots, structural dependencies etc...) and therefore comparing modified sites on different residues and in different protein regions and with non-modified residues has to be very stringently controlled.

****Points for consideration****

- Very basic question: How do you assessed the RSA value of the residues from the alphafold structure. If it is sequence based, then it is unclear what the alpha fold structure actually contributes in this step? Although I assume it is structure based, it is not well described, only a reference.

- Given that the different residues S,T,Y but also K for glycosylations etc. have a very different baseline RSA distribution, the distributions of modified residues as such are not so informative. Are the distributions of residues with the alpha fold LOD 0.65 different between modified and non-modified?

- Same point: it is very clear that "tyrosine presenting a larger proportion of cryptic phospho-sites", as they mainly are within folded domains to begin with. The pattern of phosphorylation and clustering is very different between the modified amino acid residue T,S,Y and needs consideration, given the large number of PTMs, a simple distribution is not sufficient to argue.

- Figure 3 E (proteins need names in the figure): the cryptic site T222 (Chk1) is not in the quasi ridged domain, it is in a light color region. What is actually the SPECTRUS cutoff? The Pidc is only one sentence in the main text? It says fewer than 80% intradomain contacts in rigid domains i.e. >0.8, right, but is the domain rigid?
- The evolutionary comparison (which is not my core expertise), seems again like comparing different things. Why not comparing cryptic and non-cryptic sites in the same protein regions? Also p-Y are, evolutionarily speaking, very different to p-S and p-T. How is this possibly considered in one distribution. p-Y analysis needs to be separated from the p-T and p-S analyses here.
- Have the authors thought of randomization of their data to see whether the distributions are significant?
- Labeling in Suppl Figures is insufficient. E.g. In S6 what are the various WT, A and D numbering, are this independent stable transfections/clones? Figure S7 what is R?
- Whether or not findings are "impressive" should be up to the reader, please remove these attributes in the text.

2. Significance:

Significance (Required)

The identification of buried modification sites and what the biological meaning / implications are is a very interesting topic. However PTM distribution on proteins is very skewed (many papers have identified cluster, hot spots, structural dependencies etc...) and therefore comparing modified sites on different residues and in different protein regions and with non-modified residues has to be very stringently controlled.

main conclusion: 5% of all known phospho-sites are cryptic, at least one in 1/3 of structured protein regions.

3. How much time do you estimate the authors will need to complete the suggested revisions:

Estimated time to Complete Revisions (Required)

(Decision Recommendation)

Between 1 and 3 months

4. Review Commons values the work of reviewers and encourages them to get credit for their work. Select 'Yes' below to register your reviewing activity at Web of Science Reviewer Recognition Service (formerly Publons); note that the content of your review will not be visible on Web of Science.

Yes

Review #3

1. Evidence, reproducibility and clarity:

Evidence, reproducibility and clarity (Required)

****Summary****

The methods applied in this study were thoughtfully designed. The study's goals and the experiments performed to test several of their hypotheses were meticulously planned, ensuring that the research approach was robust and aligned with the objectives. The experimental design effectively addressed the key questions and provided reliable insights into the role of cryptic PTMs in protein function and disease mechanisms.

This study investigates cryptic post-translational modification (PTM) sites in the human proteome and their role in protein folding and expression, with significant implications for disease mechanisms. This work seeks to bridge the gap between the abundance of identified PTM sites and their regulatory roles in signaling pathways. A key focus of the study is on intermediate protein conformations-states that exist between fully folded and unfolded structures to determine whether these transient states contribute to disease by affecting protein synthesis, activity, stability, and degradation. To classify PTM sites as cryptic or non-cryptic, the authors used AlphaFold-predicted structures and relative solvent accessibility (RSA) scores, excluding those within quasi-rigid domain interfaces. This enabled them to create a database of mapped PTM sites, distinguishing based on their cryptic nature. Their analysis revealed that most PTMs occur at solvent-exposed residues, but unexpectedly, one-third of tyrosine phosphosites were cryptic. To assess the impact of cryptic phosphorylation on protein expression, they performed molecular dynamics (MD) simulations on SMAD and CHK1 phosphosites, showing that cryptic sites can become transiently exposed during protein folding. Their computational simulations further supported the finding that this exposure enhancing the chances of being modified and ultimately a potential mechanism for destabilization of its structure (due to that modification) to trigger degradation in physiological conditions. Experimentally, western blotting and protein half-life measurements confirmed that phosphomimetic substitutions

affected protein expression, supporting their hypothesis that cryptic phosphorylation can influence protein stability and function. From an evolutionary and functional perspective, their phylogenetic analysis using entropy scores indicates that cryptic sites are more conserved. They also show that the cryptic PTM sites identified in this study were found to be substituted by phosphomimetic mutations in tumor-suppressor proteins, leading to dysregulation of their function and suppression of downstream signaling essential for tumor cell death. This study provides a framework for mapping cryptic PTM sites and understanding their role within intermediate protein folding states. By linking cryptic PTMs to their effects on protein stability, signaling pathways, and disease progression, the findings highlight a potential regulatory mechanism through which cryptic modifications contribute to cancer and other diseases.

****Minor revisions****

1. Result 1 - Residues with pLDDT scores below 65 were excluded from the analysis. The high-confidence measure applies to individual residues, regardless of whether the domains they belong to are also predicted with high confidence. Identifying the number of domains containing PTMs with overall high-confidence predictions could provide better insights into the orientation of modified residues within domain structures. To assess the relationship between residue-specific confidence and domain stability, we can analyze the correlation between high-confidence modified residues and the overall prediction accuracy of their domains. This could be quantified using the average error scores of domain residues. Additionally, using the average pLDDT score would indicate how many individual residues were predicted with high local structural confidence. In contrast, the average PAE (Predicted Aligned Error) score would provide insights into how well each residue's position is predicted relative to others within the domain, reflecting overall domain structural confidence.

2. "Approximately 65% of proteins with cryptic phosphosites contained only one or two such residues, while less than 10% had five or more sites (Supp. Figure 3)." To better interpret this trend, it would be useful to analyze the total number of cryptic PTMs on proteins part of this study, including all modification types-not just phosphorylation. This would help determine whether the observed pattern is specific to phosphorylation or if it extends to other post-translational modifications as well.

3. For the validation of cryptic sites, selecting domains under 200 amino acids was mentioned. However, was there also a minimum length threshold applied, similar to the filtering criteria used for false positives (less than 40 ignored)?

4. To test their hypothesis that phosphorylation affects protein expression, they selected candidates for serine and threonine but excluded tyrosine. What were the reasons for not

including tyrosine-related PTMs in their analysis?

5. Do we know that the regulatory role of S300 on PYST1 is associated with the dual specificity of the phosphatase, and is this why it was selected as a negative regulator? While the regulatory roles of the other analyzed phosphosites on SMAD and CHK1 are discussed, there is limited mention of the specific role of S300 on PYST1 within the scope of the study.

6. When comparing the entropic scores between cryptic and non-cryptic residues, the medians are 0.43 and 0.52, respectively. Although this difference is not very high, they do observe that cryptic residues have lower scores than non-cryptic ones. The distributions also show greater overlap (Figure 6). I'm wondering if any statistical testing would help assess how distinct these two groups really are.

7. Why did the authors choose to rely on AlphaFold data instead of examining PDB structures? I didn't see any explanation or rationale provided for preferring AlphaFold predictions over experimentally determined structures from the PDB.

2. Significance:

Significance (Required)

Novelty - The concept that cryptic site modifications can dysregulate signaling in cancer and other diseases is known, but systematically categorizing PTM sites into cryptic and non-cryptic to generate hypotheses for a wide range of identified PTMs remains an underdeveloped approach. This study establishes a framework for classifying PTMs based on their structural accessibility, integrating AlphaFold predictions, molecular dynamics simulations, solvent accessibility analysis, and phylogenetic conservation metrics. This approach not only enhances our understanding of PTM-mediated regulatory mechanisms but also provides a foundation for exploring how cryptic modifications contribute to protein function, stability, and disease progression.

Strengths - This study benefits from its use of multiple validation methods and false-positive filtering, resulting in a high-confidence dataset of annotated PTM sites. The combination of computational predictions and experimental analyses strengthens the validity of their findings. This integrative approach enhances the reliability of the data and provides a comprehensive understanding of cryptic versus non-cryptic PTMs in protein regulation.

Limitations

1. The study relies primarily on predicted protein structures (e.g., AlphaFold), without

exploring experimentally derived structures, which could provide more accurate and physiologically relevant insights.

2. While the research demonstrates the impact of cryptic PTMs on protein function, it would be valuable to also investigate non-cryptic sites from their annotated data. By examining the effects of modifications on these non-cryptic sites, the study could further validate the importance of the cryptic versus non-cryptic classifications and help clarify the functional relevance of both types of sites.

Audience - The broader implications of this work extend to biomedical research, drug discovery, and therapeutic development. Researchers in cell signaling and systems biology who aim to understand which modification sites are crucial for evaluating the outcomes of signaling pathways can benefit from the insights generated by this study. It provides a pathway for identifying novel drug targets and enhances our understanding of disease mechanisms, particularly in cancer and other diseases. Additionally, this work encourages and motivates computational biologists to develop more efficient methods for capturing protein folding dynamics, enabling more accurate hypotheses regarding the effects of specific PTM sites and how they influence protein function and disease progression.

My expertise lies primarily in structural biology, with a strong background in developing and utilizing bioinformatics and computational tools. While I currently have less hands-on experience with experimental techniques, my comprehensive understanding of experimental methodologies, combined with an awareness of the expected outcomes, has enabled me to effectively evaluate and interpret experimental results.

3. How much time do you estimate the authors will need to complete the suggested revisions:

Estimated time to Complete Revisions (Required)

(Decision Recommendation)

Between 1 and 3 months

Yes

Full Revision

Manuscript number: RC-2025-02892

Corresponding author(s): Emiliano, Biasini

1. General Statements

Dear Dr. Monaco,

Thank you for forwarding the reviewers' comments for our manuscript titled "*Mapping Cryptic Phosphorylation Sites in the Human Proteome*". We appreciate the thoughtful and constructive feedback, which has helped us improve the clarity and depth of our work.

We have prepared a detailed point-by-point reply addressing all the comments. In addition, we are submitting a revised version of the manuscript that includes new data and a revised text to reflect the changes made (we are also providing a file of the text highlighting these changes).

We hope the revised manuscript and our responses meet the expectations of the reviewers and the Review Commons editorial board.

Sincerely,

Emiliano Biasini
(on behalf of all co-authors)

Reviewer #1

1.1. It would be helpful if the authors could discuss whether there is any correlation between cryptic sites and the extent of experimental validation in the Phosphosite database (e.g. those that were only identified in one or a few MS experiments). It is difficult to determine stoichiometry of phosphorylation experimentally, but can any inference be made on the extent of phosphorylation of cryptic sites vs. more conventional sites located in IDRs or on the surface of globular domains?

We thank the reviewer for this valuable suggestion. To investigate the extent of the experimental validation of phosphosites, we examined the number of supporting studies for each site reported in the PhosphoSitePlus database. Specifically, we summed the values of the LT_LIT (literature-based experiments), MS_LIT (mass spectrometry literature), and MS_CST (Cell Signaling Technology mass spectrometry) fields to count the number of independent studies supporting each phosphorylation site, either cryptic or non-cryptic. To visualize the results, we plotted the number of supporting references vs the relative solvent accessibility (RSA) distribution of phosphosites (Figure R1). The analysis revealed a direct correlation between the RSA of phosphosites and the number of studies supporting their phosphorylation. This observation may arise from an intrinsic difficulty in studying cryptic phosphosites due to their destabilizing effects on native proteins. Notably, no differences were observed in the number of supporting studies within cryptic phosphosites (Figure R1B). We have not mentioned these analyses in the new version of the manuscript. However, we would gladly add it if the editor or the reviewer advises accordingly.

Figure R1. A. RSA distribution of phosphosites colored for the number of studies supporting their phosphorylation. **B.** RSA distribution of cryptic phosphosites colored for the number of studies supporting their phosphorylation. References are intended as the sum of LT_LIT (Literature matches) MS_LIT (MS experiment match) and MS_CST values from the PhosphositePlus dataset.

Full Revision

1.2. The authors note that a larger percentage of tyrosine phosphorylation sites are cryptic compared with serine/threonine sites. I assume that tyrosine itself is more highly enriched in the hydrophobic cores of proteins relative to serine or threonine, due to its bulky hydrophobic side chain. Is the increased proportion of cryptic tyrosine phosphorylation sites more, less, or the same as the proportion of tyrosine in hydrophobic cores relative to serine and threonine?

We thank the reviewer for this insightful comment. As correctly noted, tyrosine residues tend to be enriched in the hydrophobic cores of proteins, as reflected by their generally lower relative solvent accessibility (RSA) values, regardless of phosphorylation state. This enrichment is likely due to the tyrosine side chain's bulky and partially hydrophobic nature. To address the reviewer's question, we compared the RSA distributions of phosphorylated tyrosine, serine, and threonine residues with that of the same residues non-phosphorylated in the human proteome (Figure R2). In order to statistically compare the two distributions, we employed the Mann-Whitney test. The large sample size inevitably yields very low p-values, even when the distributions differ mildly (pThr, pSer vs non-p Thr, Ser, $p < 0.00001$; pTyr vs non-p Tyr, $p < 0.00001$). Therefore, we also calculated the Cohen's d value, which quantifies the effect size difference between the means of two groups in terms of standard deviations (pThr, pSer vs non-p Thr, Ser, Cohen's $d = 0.180$; pTyr vs non-p Tyr, Cohen's $d = 0.161$). These data indicate that while the RSA distributions are significantly different, the extent of such difference reflects a minor effect size. Therefore, the proportion of cryptic tyrosine, serine, and threonine phosphorylation sites is similar to that of the same residues in hydrophobic cores of proteins. Still, their existence represents a notable observation since phosphorylation is easily predicted to cause destabilizing effects when occurring in protein cores. While we have not included these analyses in the new version of the manuscript, we would be pleased to add them if the editor or the reviewer advises us accordingly.

Figure R2. A. RSA kernel density estimation of phosphorylated tyrosine sites (purple) compared to non-phosphorylated tyrosine residues in the human proteome (gray). **B.** RSA kernel density estimation of phosphorylated serine and threonine sites (purple) compared to non-phosphorylated serine and threonine residues in the human proteome (gray).

1.3. Fig. 5D and E: I had some trouble interpreting these figures. Indicating where the native state is in the plots would be helpful (stated in text as lower right, but a rectangle on the plot would make this more obvious). The text discusses three metastable intermediates, but what is the fourth one shown on the figures (well A, close to the native state)? This could be more explicitly explained.

We added the missing rectangles into the original Fig. 5D and E (see below Figure R3 and R4). The three metastable intermediates discussed in the original text reflect protein conformers in which the cryptic site is exposed to the solvent. Conversely, the fourth state, and the final native state, are conformations in which the site is already partially or fully cryptic. The observation that the masking of cryptic sites coincides with the latest folding steps allows us to hypothesize a mechanism by which cryptic phosphorylation may regulate protein folding. Following the reviewer's suggestion, we now specify more explicitly each conformation in the new version of the legends of the relative figures (text file with track changes, lines 950 and 1017).

Figure R3. D. Atomistic reconstruction of the CHK1 KA1 domain folding pathway. Lower-bound approximation of the transition path energy is related to the folding of KA1. The energy is plotted as the negative logarithm of the probability distribution ($-\ln(p)$), which is expressed as a function of the collective variables Q and RMSD (65x65 bin matrix). Gaussian blur was applied. The highly populated native state appears as expected in the bottom-right corner (high Q and low RMSD, black rectangle). The indexed squares define the most-populated partially unfolded regions of interest ($-\ln(p)$). Well thresholds from most to least stable: $A \leq 2.5$ k. bT, $B \leq 3.5$ kbT, $C \leq 3.5$ kbT, $D \leq 3$ kbT). **E.** Distribution of RSA of amino acid T382 along the transition path. The highest values are associated with the unfolded state (top left), while in the native state the residue is consistently below 0.15. **F.** Representative conformations for each cluster. All clusters are explored at least by 2/9 LB conformations. Residue T382 is displayed in red, α -helices are colored in purple, β -sheets in orange.

Figure R4. B. The plot shows the atomistic reconstruction of the SMAD2 folding pathway. Lower-bound approximation of the transition path energy is related to the folding of SMAD2. The energy is plotted as the negative logarithm of the probability distribution ($-\ln(p)$), which is expressed as a function of the collective variables Q and RMSD (65x65 bin matrix). Gaussian blur was applied. The highly populated native state appears as expected in the bottom-right corner (high Q and low RMSD, black rectangle). The indexed squares define the most-populated partially unfolded regions of interest ($-\ln(p)$). **C.** The plot shows the distribution of RSA of the S417 residue along the transition path. The highest values are associated with the unfolded state (top left), while in the native state the RSA is consistently below 0.15. **D.** Representative conformations for each cluster. S417 is displayed in red, α -helices are colored in orange, β -sheets in light purple.

Full Revision

1.4. The fact that phosphomimetic mutations of cryptic sites in SMAD2 and CHK1 lead to lower expression levels and shorter half-lives is not surprising, given the expected disruption of the hydrophobic core by introduction of a charged residue. The results certainly show that if phosphorylated, these sites would decrease expression and half-life. With respect to half-life, however, if the authors are correct and cryptic sites are predominately phosphorylated co-translationally, one would expect that the half-life curves for the wt protein would not be a simple exponential, but would instead reflect two distinct populations: those that are phosphorylated during translation, and are almost immediately degraded, and those that escape phosphorylation and have the same half-life as the non-phosphorylatable mutant. Are the actual experimental results consistent with this two-population model? If not, this would be evidence that some of these cryptic sites can be exposed post-translation, either by thermal fluctuation or biological interactions.

We thank the reviewer for this insightful point. The readout employed in our study (i.e., western blotting) measures the aggregate signal from the total protein population in the cell culture. It thus reflects average protein levels rather than the dynamics of individual molecules. As such, it is not well-suited to resolving coexisting populations with distinct half-lives. We agree that if phosphorylation of cryptic sites occurs strictly co-translationally, one might expect a biphasic decay curve. However, due to methodological constraints, our assay provides only a single exponential fit to the global turnover kinetics. While we cannot entirely exclude the possibility that cryptic sites may become exposed post-translationally (e.g., due to thermal fluctuations or interactions), our molecular dynamics simulations did not reveal such exposure events within the simulated timescales. Therefore, while the two-population model remains plausible in principle, our results are consistent with a co-translational phosphorylation and degradation model. Forthcoming experiments aimed at characterizing the phosphorylation of ribosome-associated nascent chains in the human proteome may further validate this conclusion.

1.5. The authors make a point that cryptic phosphosites are more highly conserved than non-cryptic phosphosites, but it is not clear to me whether it is the side chain itself or its ability to be phosphorylated that is conserved. Supplemental Fig. 9, if I am interpreting it correctly, would suggest it is the residue itself and not its phosphorylation that is conserved. If so, wouldn't this suggest that phosphorylation of these cryptic sites is just an inevitable consequence of the conservation of serine, threonine, and tyrosine residues in hydrophobic core regions? If the authors have evidence that argues against this simple hypothesis, they should discuss it (e.g., cryptic phosphosites are more highly conserved in some cases than non-phosphorylated tyrosine, serine, and threonine residues that are not solvent accessible).

We agree with the reviewer's interpretation. The higher conservation of cryptic phosphosites likely reflects the evolutionary constraint on hydrophobic core residues, which tend to be more conserved due to their role in structural stability. This conservation does not imply phosphorylation at those sites is functionally selected across species. Instead, when such residues are phosphorylated, as we observe in the human proteome, the effect is often destabilizing and associated with protein degradation. Our analysis does not establish that the phosphorylation of cryptic residues is conserved across species, only that the residues themselves are. We appreciate the reviewer's suggestion and now explicitly discuss this point in the revised

Full Revision

manuscript to clarify the distinction between residue conservation and phosphorylation conservation (text file with track changes, line 618)

1.6. Regarding the evolutionary conservation of cryptic sites, have the authors taken into consideration that tyrosine-specific kinases, phosphatases, and reader domains first appeared in the first metazoans, and are for the most part not seen in non-metazoan eukaryotes? I notice some of the proteomes used for the conservation analysis include plants and yeast, which lack most tyrosine phosphorylation.

We thank the reviewer for this insightful comment. In response to the suggestion, we have recalculated the entropic conservation score by restricting the analysis to metazoan species. This analysis ensures that the evolutionary context more accurately reflects the presence and functional relevance of tyrosine-specific kinases, phosphatases, and reader domains. The comparison between the entropic score distribution calculated by including or not non-metazoan orthologues show statistically significant differences for both serine and threonine, and tyrosine. However, the large sample sizes translate inevitably into statistically significant p-values, even when the differences in mean are minimal and the standard deviations relatively small. To better assess the practical relevance of these differences, we calculated Cohen's *d* as a measure of effect size (Table R1). The coefficient helps assess the size and biological significance of a difference (>0.2 = small effect; >0.5 = medium effect; >0.8 = large effect). The analysis indicates a very modest deviation in entropic scores by including or not non-metazoan orthologues.

Table R1	ES Serine & Threonine (+/- non-metazoan)	ES Tyrosine (+/- non-metazoan)
Mann-Whitney (p value)	>0.00001	0.0001
Cohen's d	0.1037	0.0858

Figure R5. A. Entropic score kernel density estimation of cryptic serine and threonine phosphorylation sites including (light purple) or not (purple) non-metazoan orthologues. **B.** Entropic score kernel density estimation of cryptic tyrosine phosphorylation sites including (light purple) or not (purple) non-metazoan orthologues. The Whitney *p* value is set to zero when estimated to be below 10^{-5} .

1.7. I find the argument that phosphorylation of exposed core residues is part of normal protein quality control/proteostasis to be convincing. Can the authors provide any experimental evidence to support this model (for example, greater phosphorylation of cryptic sites under stress conditions)? I don't think these experiments are necessary, but would seem to be a logical next step and could be done quite easily through collaboration.

We appreciate the reviewer's suggestion and fully agree that showing more significant phosphorylation of cryptic sites under stress conditions could represent an exciting future direction. We are conducting experiments on individual tumor suppressors such as p53 and PTEN, which harbor cryptic phosphosites, to test whether cellular stress conditions enhance phosphorylation at these positions. These studies assess whether such modifications contribute to altered protein stability or function in stress or disease contexts, particularly cancer. We plan to communicate these results in forthcoming publications and are currently open to collaborations to broaden this line of investigation.

1.8. The authors note at the end of the discussion that targeting cryptic phosphosites might be a strategy to selectively degrade some proteins in cancer. Practically, how would this work? I can't think of how, but perhaps the authors can provide more specific suggestions.

We thank the reviewer for raising this important point. One promising approach to therapeutically exploit cryptic phosphosites builds on the PPI-FIT principles (Pharmacological Protein Inactivation by Folding Intermediate Targeting). This strategy targets transient structural pockets appearing only in folding intermediates (Spagnolli et al., *Comm Biology* 2021). In this context, kinases that phosphorylate cryptic sites could be modulated, either inhibited or redirected, so that misfolded or oncogenic proteins are selectively marked for degradation. For example, selectively enhancing the phosphorylation of a cryptic site on an oncogenic protein could destabilize it and promote its degradation via the proteasome. Conversely, preventing phosphorylation at a cryptic site on a tumor suppressor (e.g., by inhibiting the specific kinase) could enhance protein stability and restore function. While this concept is still emerging, it offers an exciting therapeutic avenue that complements our findings. We added a paragraph addressing this point in the discussion section of the new version of the manuscript (text file with track changes, line 716).

1.9. Introduction: "It involves the addition of a phosphate to an hydroxyl group found in the side chain of specific amino acids, typically serine, threonine or tyrosine residues." Of course serine, threonine, and tyrosine are the only standard amino acids with a simple hydroxyl group, so "typically" is not needed here.

We have removed the word "typically" to reflect the accurate chemical specificity of phosphorylation events (text file with track changes, line 82).

1.10. In my view this is an important study, bringing rigor and a broad proteomic perspective to a phenomenon that (to my knowledge) had not been carefully examined previously. In terms of the big picture, I am of two minds. On the one hand, showing that phosphorylation of hydrophobic core residues exposed during translation or the early stages of folding can regulate steady state levels of some proteins provides an intriguing

Full Revision

new mechanism to control the complement of proteins in the cell, and is potentially an area of regulation in normal physiology or in disease. On the other hand, if this is just part of the normal proteostatic mechanisms (hydrophobic core residues exposed for too long consign the protein to degradation, before it can lead to aggregation and other problems), that is a little less interesting to me. I think future work to tease out whether this mechanism is actually regulated and used by the cell to transmit information will be key. But the first step is showing that the phenomenon is real and widespread, and in my view this preprint accomplishes that goal very well.

We appreciate the reviewer's thoughtful summary and agree that distinguishing between passive proteostatic clearance and active regulatory function is essential. Toward this goal, we plan to carry out a phosphoproteomic analysis of ribosome-associated nascent chains. By mapping phosphorylation events during translation, we aim to validate our cryptic phosphosite dataset in a co-translational context and potentially identify novel regulatory modifications. This approach will also help us assess whether phosphorylation at cryptic sites is modulated context-dependently, thereby supporting a role in regulated protein expression rather than solely quality control.

Reviewer #2

2.1. Evolutionary comparison whether cryptic and non-cryptic sites are differently conserved. Two distinct distributions for cryptic and non-cryptic phospho-sites are observed and Figure 6 shows two entropy distributions of cryptic v non-cryptic. Here it is unclear whether this is significant given the different distributions of the two types when non modified.

We thank the reviewer for raising this critical point. Due to the large sample sizes in our analysis, statistical tests inevitably yield very low p-values, even when differences in mean are minimal and the standard deviations relatively small. To better assess the practical relevance of these differences, we calculated Cohen's *d* as a measure of effect size (Table R2). The comparison between cryptic and non-cryptic phosphosites yielded an effect size (Cohen's *d* = 0.4028) slightly lower than the one obtained for residues lying within protein cores or exposed on protein surfaces (Cohen's *d* = 0.5126), both indicating a modest but meaningful shift in entropic scores. In contrast, the comparisons between cryptic phosphosites and all core residues, as well as non-cryptic phosphosites and all surface residues, showed negligible effect sizes (Cohen's *d* = 0.0245 and 0.1326, respectively). These findings suggest that while statistical significance is achieved in all cases, only the difference between cryptic and non-cryptic phosphosites, or core and surface residues, reflects a meaningful biological signal. We have now included these data in the new version of the manuscript (text file with track changes, line 544)

Table R2	Cryptic vs Non-Cryptic phosphosites	Core vs Surface Residues	Cryptic phosphosites vs core residues	Non-cryptic phosphosites vs surface residues
Mann-Whitney (p value)	<0.00001	<0.00001	0.0152	<0.00001
Cohen's d	0.4028	0.5126	0.0245	0.1326

2.2. The identification of buried modification sites and what the biological meaning / implications are is a very interesting topic. However PTM distribution on proteins is very skewed (many papers have identified clusters, hot spots, structural dependencies etc...) and therefore comparing modified sites on different residues and in different protein regions and with non-modified residues has to be very stringently controlled.

We fully agree with the reviewer that PTM distribution is non-random and influenced by structural and functional constraints, making comparative analyses challenging. To ensure rigor, we implemented a robust computational pipeline. Unlike other PTMs found almost exclusively on solvent-exposed residues, phosphorylation uniquely showed a distinct subset of sites with

Full Revision

extremely low solvent accessibility. This pattern held even after applying stringent structural and dynamical filters. Specifically, we excluded low-confidence residues, small or unstructured domains, and sites that become exposed due to thermal fluctuations, using the SPECTRUS-based dynamic analysis. While we cannot entirely rule out context-specific exposure in fully folded proteins (e.g., during protein-protein interactions), we validated selected cryptic sites experimentally, and our findings were consistent with the computational predictions. We believe this multilayered approach strengthens the reliability of our classification and distinguishes cryptic phosphosites from the broader PTM landscape.

2.3. Very basic question: How do you assessed the RSA value of the residues from the alphafold structure. If it is sequence based, then it is unclear what the alpha fold structure actually contributes in this step? Although I assume it is structure based, it is not well described, only a reference.

We calculated the RSA values using the Shrake-Rupley algorithm implemented in the MDTraj Python library. This is a structure-based metric: for each PTM-carrying residue, we evaluated the absolute SASA from the 3D AlphaFold structure and normalized it against the theoretical maximum exposure for that residue in a Gly-X-Gly tripeptide, as defined in Tien et al. (2013). Thus, AlphaFold structures directly provide the atomic coordinates necessary for solvent accessibility estimation. We have now revised the Methods section to describe this process more explicitly (text file with track changes, lines 110 and 113).

2.4. Given that the different residues S,T,Y but also K for glycosylations etc. have a very different baseline RSA distribution, the distributions of modified residues as such are not so informative. Are the distributions of residues with the alpha fold LOD 0.65 different between modified and non-modified?

2.5. Same point: it is very clear that "tyrosine presenting a larger proportion of cryptic phosphor-sites", as they mainly are within folded domains to begin with. The pattern of phosphorylation and clustering is very different between the modified amino acid residue T,S,Y and needs consideration, given the large number of PTMs, a simple distribution is not sufficient to argue.

As already discussed in point 1.2 above, and correctly noted also by this reviewer, tyrosine residues are generally enriched in the hydrophobic cores of proteins, which is reflected by their typically low RSA, regardless of phosphorylation status. This tendency likely arises from the bulky and partially hydrophobic nature of the tyrosine side chain. To address the reviewer's question, we compared the RSA distributions of phosphorylated tyrosine, serine, and threonine residues with those of all these amino acids in the human proteome. We found that phosphorylated residues consistently exhibit higher RSA values than the overall averages for their respective amino acids. This is expected, as phosphorylation within protein cores would likely be destabilizing. Indeed, the existence of low-RSA phosphorylated residues, represents a significant deviation from the intrinsic tendency of tyrosine, serine, and threonine residues and suggests that cryptic sites may become accessible only transiently along protein folding pathways.

2.6. Figure 3E (proteins need names in the figure): the cryptic site T222 (Chk1) is not in the quasi ridged domain, it is in a light color region. What is actually the SPECTRUS cutoff?

Full Revision

The Pidc is only one sentence in the main text? It says fewer than 80% intradomain contacts in rigid domains i.e. >0.8, right, but is the domain rigid?

We have revised the original figure in the new version of the manuscript to include protein names, and clarified the domain assignments. The cryptic phosphosite T222 in Chk1 lies within a quasi-rigid domain, as identified by SPECTRUS. The color of the image does not reflect any structural property but instead it is used to distinguish different quasi-rigid domains. In particular, black regions identify unstructured domains, whereas shadows from dark grey to white identify quasi-rigid domains. We apologize for the lack of clarity. We have corrected the figure legend accordingly (text file with track changes, line 912).

There is no cutoff in SPECTRUS' identification of quasi-rigid domain. Non quasi-rigid domains are simply regions of the protein that SPECTRUS cannot process properly. Meaning regions that, due to the large degree of intrinsic fluctuations, cannot be modelled as quasi-rigid.

We also expanded the description of Pidc in the main text to clarify that it quantifies the proportion of intra-domain contacts made by the phosphosite's side chain, and that a cutoff of ≥ 0.8 was used to retain only residues well-integrated within rigid domains (text file with track changes, line 243). We hope these updates will resolve the ambiguities noted and more clearly define the criteria used in our filtering pipeline.

2.7. The evolutionary comparison (which is not my core expertise), seems again like comparing different things. Why not comparing cryptic and non-cryptic sites in the same protein regions? Also p-Y are, evolutionarily speaking, very different to p-S and p-T. How is this possibly considered in one distribution. p-Y analysis needs to be separated from the p-T and p-S analyses here.

We want to clarify that our evolutionary analyses compare residues at the aligned positions in orthologous proteins across multiple species. This approach ensures that each cryptic or non-cryptic phosphosite is assessed in its native structural and sequence context. Therefore, the comparison is not between different regions but evaluates the evolutionary conservation of specific sites across species, allowing for a direct and meaningful comparison of cryptic and non-cryptic phosphosites. In order to address the second point, we report below the entropic score distributions for serine/threonine and tyrosine, separately (Figure R5).

Figure R5. A. Entropic score kernel density estimation of cryptic and non-cryptic serine (S) and threonine (T) phosphorylation sites. Mann-Whitney U test p value:<0.0001. Cohen's d: 0.4070 (small difference). **B.** Entropic score kernel density estimation of cryptic and non-cryptic tyrosine (Y) phosphorylation sites. Mann-Whitney U test p value:<0.0001. Cohen's d: 0.3154 (small difference).

Full Revision

2.8. Have the authors thought of randomization of their data to see whether the distributions are significant?

We are unsure we fully understand what the referee means by randomizing the data in this case. However, according to the mathematical definition of entropic score, the limit case in which, within each orthogroup, the phosphorylated amino acid is replaced by a completely random residue yields an entropic score of 1. The opposite limit, in which all members of the orthogroups have the same amino acid in the position of the phosphorylated amino acid, yields an ES of 0. We have added a paragraph in the methods to stress this point (text file with track changes, line 354).

2.9. Labeling in Suppl Figures is insufficient. E.g. In S6 what are the various WT, A and D numbering, are this independent stable transfections/clones? Figure S7 what is R?

Thank you for pointing this out. We have now corrected the missing information in the revised version of the manuscript (text file with track changes, from line 992 to 1008)

2.10. Whether or not findings are "impressive" should be up to the reader, please remove these attributes in the text.

We agree with the reviewer's suggestion. We have removed subjective language such as "impressive" from the revised manuscript to ensure an objective and neutral tone, allowing readers to independently evaluate the significance of our findings (text file with track changes, line 454).

Reviewer #3

3.1. Residues with pLDDT scores below 65 were excluded from the analysis. The high-confidence measure applies to individual residues, regardless of whether the domains they belong to are also predicted with high confidence. Identifying the number of domains containing PTMs with overall high-confidence predictions could provide better insights into the orientation of modified residues within domain structures. To assess the relationship between residue-specific confidence and domain stability, we can analyze the correlation between high-confidence modified residues and the overall prediction accuracy of their domains. This could be quantified using the average error scores of domain residues. Additionally, using the average pLDDT score would indicate how many individual residues were predicted with high local structural confidence. In contrast, the average PAE (Predicted Aligned Error) score would provide insights into how well each residue's position is predicted relative to others within the domain, reflecting overall domain structural confidence.

Our analysis excluded residues with pLDDT scores below 65 to ensure high local confidence. While pLDDT provides residue-level structural confidence, assessing domain-wide prediction quality offers additional insights into modified residues' spatial organization and exposure. However, a domain-level interpretation is currently limited by the format of AlphaFold structural predictions. Specifically, AlphaFold does not provide Predicted Aligned Error (PAE) matrices for sequences split into overlapping fragments, a method used for proteins longer than 2,700 amino acids. These fragment predictions are only available in the downloadable AlphaFold proteome archives, not through the web interface, and lack the global alignment metrics (such as PAE) necessary for analyzing domain stability or inter-residue confidence within the domain context.

3.2. "Approximately 65% of proteins with cryptic phosphosites contained only one or two such residues, while less than 10% had five or more sites (Supp. Figure 3)." To better interpret this trend, it would be useful to analyze the total number of cryptic PTMs on proteins part of this study, including all modification types-not just phosphorylation. This would help determine whether the observed pattern is specific to phosphorylation or if it extends to other post-translational modifications as well.

To compare the occurrence of different cryptic PTMs, we extended our analysis to include all cryptic post-translational modifications annotated in PhosphoSitePlus, including phosphorylation, glycosylation, methylation, sumoylation, and ubiquitination. The approach allowed us to assess whether the observed distribution of cryptic phosphosites is unique or represents a more general feature of all cryptic PTMs. We observed extensive variation among the different PTMs in the proportion of proteins carrying 1, 2, or more of the same cryptic PTM (see Table R3). However, it must be noted that the relatively low number of cryptic PTMs, excluding phosphorylation, could make it difficult to determine whether these patterns reflect actual biological trends or are simply influenced by the sample size. We have not included these data in the new version of the manuscript, but we would be willing to add them if the editor or the reviewer advises us accordingly.

Table R3	Putative Cryptic Sites*	n. of Proteins Containing at least 1 Cryptic PTM	Proteins with 1 putative cryptic (%)	Proteins with 2 putative cryptic (%)	Proteins with ≥3 putative cryptic (%)
Phosphorylation	23118	7,478	3643 (49%)	1972 (26%)	1863 (25%)
Ubiquitination	3708	2,121	1642 (77%)	468 (22%)	111 (5%)
Acetylation	627	550	509 (93%)	38 (7%)	3 (>1%)
Methylation	468	387	335 (87%)	47 (12%)	5 (1%)
Sumoylation	127	110	99 (90%)	11 (10%)	0 (0%)
Glycosylation	66	47	40 (85%)	6 (13%)	1 (2%)

* Cryptic sites predicted without applying the SPECTRUS-based dynamic filtering

3.3. For the validation of cryptic sites, selecting domains under 200 amino acids was mentioned. However, was there also a minimum length threshold applied, similar to the filtering criteria used for false positives (less than 40 ignored)?

The 40-residue threshold was applied because protein domains that are too small cannot be reliably subdivided into quasi-rigid domains. Trying to run SPECTRUS on structures with fewer than 40 residues inevitably returns a warning, reflecting the intrinsic cooperative nature of quasi-rigid domains. In fact, entities composed of too few amino acids cannot properly arrange themselves into 3D structures and tend to be disordered. The same reasoning was applied when choosing the proteins to simulate. In particular, for the refolding simulations, we selected protein domains possessing the following properties:

1. Shorter than 200 amino acids to limit the computational demands.
2. Long enough to fold into an ordered 3-dimensional conformation reliably.
3. Have an experimentally determined NMR or X-ray crystal structure

3.4. To test their hypothesis that phosphorylation affects protein expression, they selected candidates for serine and threonine but excluded tyrosine. What were the reasons for not including tyrosine-related PTMs in their analysis?

Our experimental assays relied on phosphomimetic substitutions to mimic the effect of phosphorylation. While serine/threonine phosphorylation can be reasonably mimicked by E or D substitutions, there is no reliable single-residue mimic for phosphotyrosine. Indeed, E or D substitutions do not recapitulate the structural or electronic features of pTyr. Given these limitations, we excluded tyrosine phosphosites from experimental validation to avoid generating inconclusive or misleading data.

3.5. Do we know that the regulatory role of S300 on PYST1 is associated with the dual specificity of the phosphatase, and is this why it was selected as a negative regulator? While the regulatory roles of the other analyzed phosphosites on SMAD and CHK1 are discussed, there is limited mention of the specific role of S300 on PYST1 within the scope of the study.

S300 of PYST1 was selected not due to known regulatory relevance, but for technical convenience. PYST1 is a relatively small protein, facilitating computational simulations. We also had suitable reagents for detection (i.e., expression vector), and importantly, S300 was identified as a false-positive cryptic phosphosite removed by our dynamic filtering. It was a practical and structurally matched negative control for validating our computational pipeline.

3.6. When comparing the entropic scores between cryptic and non-cryptic residues, the medians are 0.43 and 0.52, respectively. Although this difference is not very high, they do observe that cryptic residues have lower scores than non-cryptic ones. The distributions also show greater overlap (Figure 6). I'm wondering if any statistical testing would help assess how distinct these two groups really are.

We thank the reviewer for the comment raised by reviewer #2, for which we provide an answer above. Briefly, given our large sample sizes, statistical tests often yield very low p-values even for minor differences. To assess the biological significance, we calculated Cohen's d (Table R2 above). The effect size between cryptic and non-cryptic phosphosites ($d = 0.4028$) was modest but meaningful, and slightly lower than between core and surface residues ($d = 0.5126$).

3.7. Why did the authors choose to rely on AlphaFold data instead of examining PDB structures? I didn't see any explanation or rationale provided for preferring AlphaFold predictions over experimentally determined structures from the PDB.

We appreciate the value of this comment. We focused on AlphaFold to maximize proteome-wide coverage. Indeed, although PDB structures offer experimentally validated conformations, their sparse and uneven proteome coverage (particularly for membrane proteins, low-abundance factors, and intrinsically disordered regions) precludes a truly global analysis. AlphaFold2 models, by contrast, deliver accurate, full-length structures for nearly the entire human proteome, enabling unbiased, large-scale mapping of cryptic phosphosites. Nonetheless, we performed the same analysis using high-resolution structures from the Protein Data Bank (PDB). The results were fully consistent with those based on AlphaFold predictions, indicating that our findings are consistent across the two databases (see Figure R6 below).

Figure R6. RSA kernel density estimation of phosphorylated tyrosine, serine, and threonine sites (purple) compared to all phosphorylated tyrosine, serine, and threonine residues in the human proteome. We collected 754 non-redundant molecular structures with 1199 eligible phosphosites. A total of 246 residues were below the RSA threshold (0.15) needed to be classified as putative cryptic sites, accounting for ~20% of the total.

3.8. Novelty - The concept that cryptic site modifications can dysregulate signaling in cancer and other diseases is known, but systematically categorizing PTM sites into cryptic and non-cryptic to generate hypotheses for a wide range of identified PTMs remains an underdeveloped approach. This study establishes a framework for classifying PTMs based on their structural accessibility, integrating AlphaFold predictions, molecular dynamics simulations, solvent accessibility analysis, and phylogenetic conservation metrics. This approach not only enhances our understanding of PTM-mediated regulatory mechanisms but also provides a foundation for exploring how cryptic modifications contribute to protein function, stability, and disease progression.

We appreciate the reviewer's comment. To our knowledge, this is the first study to introduce and define "cryptic phosphosites" as a structurally distinct and functionally relevant subset of phosphorylation sites. While some individual cases of buried amino acids influencing cancer-related proteins have been reported, no previous study has systematically mapped, filtered, and analyzed these sites across the human proteome using integrated structural, dynamical, evolutionary, and experimental criteria.

3.9. The study relies primarily on predicted protein structures (e.g., AlphaFold), without exploring experimentally derived structures, which could provide more accurate and physiologically relevant insights.

We have addressed this point above (see reply to #3.7).

3.10. While the research demonstrates the impact of cryptic PTMs on protein function, it would be valuable to also investigate non-cryptic sites from their annotated data. By examining the effects of modifications on these non-cryptic sites, the study could further

Full Revision

validate the importance of the cryptic versus non-cryptic classifications and help clarify the functional relevance of both types of sites.

We thank the referee for this thoughtful suggestion. We compared the proportion of cryptic or non-cryptic phosphosites associated with cancer- and disease-related mutations in each group from the COSMIC and PTMVar datasets. The percentage of phosphosites associated with the two repositories is essentially the same for cryptic and non-cryptic sites. This observation suggests that, despite their different structural and regulatory features, both site types occur similarly in disease contexts (see Table R4). We have included these data in the new version of the manuscript (text file with track changes, line 1067; and new Supp. Table 3).

Table R4	n. Entries	Total n. Phosphosites	Frequency
COSMIC			
Non-Cryptic	4055	207225	1.96%
Cryptic	204	10606	1.92%
PTMVar			
Non-cryptic	1270	207225	0.61%
Cryptic	138	10606	1.30%

Prof. Emiliano Biasini
University of Trento, Via Sommarive 9, 38121 Trento
Department CIBIO
Italy

11th Jul 2025

Re: EMBOJ-2025-121567-T
Mapping Cryptic Phosphorylation Sites in the Human Proteome

Dear Dr. Biasini,

Thank you for submitting your revised Review Commons manuscript for consideration by The EMBO Journal. Given the interest of the topic and the generally supportive transferred referee reports, I decided to treat the work like a regular EMBO Journal revision, and returned it directly to the original referees. All of them have now assessed the revised manuscript (see comments below), and are overall satisfied with the revisions and responses. Following incorporation of various final presentational changes still requested in the reports, we would therefore be happy to offer publication of this work in our journal.

In addition to the remaining referee points, please also take care of the following editorial issues, mainly to adjust the manuscript format according to EMBO Journal guidelines:

GENERAL:

- Please download and complete our author checklist (link provided below).
- Since we mandate addition of ORCID identifiers for all (co-)corresponding authors, please encourage Dr. Faccioli to obtain and add an ORCID to his author profile in our submission system. This needs to be added by Dr. Faccioli personally and can unfortunately not be done by others on his behalf.
- Please provide suggestions for a short 'blurb' text prefacing and summing up the conceptual aspect of the study in two sentences (max. 250 characters), followed by 3-5 one-sentence 'bullet points' with brief factual statements of key results of the paper; they will form the basis of an editor-written 'Synopsis' accompanying the online version of the article. Please also upload a synopsis image, which can be used as a "visual title" for the synopsis section of your paper. The image should be in PNG or JPG format, and please make sure that it remains in the modest dimensions of (exactly) 550 pixels wide and 300-600 pixels high.

TEXT:

- Please adjust the order of the manuscript sections: Title page with complete author information, Abstract, Keywords, Introduction, Results, Discussion, Methods, Data Availability, Acknowledgements, Disclosure and Competing Interests Statement, References, Main Figure Legends, Tables, Expanded Figure Legends.
- On the abstract page of the manuscript, please include 4-5 general keyword terms to enhance searchability.
- Please note that Materials and Methods need to be described in the main text using our 'Structured Methods' format (for detail, see <https://www.embopress.org/page/journal/14693178/authorguide#structuredmethods>). The in-text "Methods" section should contain method and protocol descriptions (ideally using a step-by-step protocol format to facilitate adoption of the methodologies across labs), while all key reagents, experimental models, software and relevant equipment - including their sources and relevant identifiers - should be listed in a separately uploaded Reagents and Tools Table, a template for which can be downloaded from the above section of our Author Guidelines.
- As we are switching from a free-text author contribution statement towards a more formal statement based on Contributor Role Taxonomy (CRediT) terms, please remove the present Author Contribution section and instead specify each author's contribution(s) directly in the Author Information page of our submission system during upload of the final manuscript. See <https://casrai.org/credit/> for more information.
- Please rename the Competing Interest section into "Disclosure and Competing Interests Statement", in accordance with our updated Guide to Authors (<https://www.embopress.org/competing-interests>)
- Please adjust the format of the reference list and of the in-text citations according to EMBO Journal format (alphabetical order,

author name et al + year, first up to 10 authors should be listed, followed by 'et al'; complete volume and page numbers - DOI info only in case of advance publications that do not yet have volume/page numbers).

- Please move the Code Availability information into the Data Availability section, it should not have a separate section heading.
- Please make sure to include all relevant funding information, both in the manuscript and in our submission system.

DATA:

- Please refer to our author guide (www.embopress.org/page/journal/14602075/authorguide#expandedview) regarding "supplementary information", and consider re-organizing the current figures and supplemental figures. We are not limited to 6 main figures, and in addition, we can have several "Expanded View" figures, whose legends would also need to be in the main text, and which would be type-set and directly visible (expandable) with the HTML version of the paper (naming/in-text callouts: Figure EV1/2-3...). Additionally/alternatively, "supplementary" content that is of comparably lesser importance may be provided in a single "Appendix" PDF, which may combine Appendix Figures, Appendix Tables, as well as Appendix Methods. Such an Appendix should be prefaced by a Table of Contents, and legends should be included next to each Appendix figure, whose naming should be adjusted to "Appendix Figure S1/2/3..." (please refer to the above-referenced section of our author guidelines for detailed information).
- Please convert the "supplementary tables" into Expanded View tables (call-out: "Table EV1/2/3..."), uploading them as individual XLSX files, with their respective legends included in a separate "legend" tab. The two shorter ones ("Suppl table" 3 & 6) might also become Appendix Tables, in which case they should appear in the Appendix PDF, and numbered separately (i.e., "Appendix Table S1/2")
- All main (and any EV figures) should be uploaded as individual files with sufficient resolution/quality for production. Please also make sure to call-out all of the individual Figure panel at least once in the text (e.g. Fig 5F currently appears not to be referenced anywhere).
- During routine pre-acceptance checks, our data editors have raised the following queries regarding figures, data, and legends; I would appreciate if you briefly answered to them in the cover letter of your final submission, and made the requested text modifications with changes/additions highlighted via the "Track changes" option, to facilitate our final checking:
 - 1) Please note that the box plots need to be defined in terms of minima, maxima, centre, bounds of box and whiskers, and percentile in the legends of figures 4A-C
 - 2) Please note that information related to N is missing in the legends of figures 4A-C
- Finally, you shall also receive a separate message from our Source Data curation team, with instructions on how to prepare and upload relevant image and numerical raw data.

Should you need additional guidance/feedback regarding this final adjustments, please do not hesitate to contact us directly. Thank you again for the opportunity to consider this work for The EMBO Journal, and I look forward to receiving your re-revised manuscript.

Yours sincerely,

Revision to The EMBO Journal should be submitted online within 90 days, unless an extension has been requested and approved by the editor; please click on the link below to submit the revision online before 9th Oct 2025:

Link Not Available

Referee #1:

This study categorizes cryptic and non-cryptic post-translational modifications (PTMs) and investigates their roles in regulating protein activity and signaling pathways. By integrating AlphaFold-predicted structures, solvent accessibility data, and stringent criteria to reduce false positives, the authors systematically identified and validated cryptic PTM sites. Through molecular dynamics (MD) simulations, they demonstrated that cryptic phosphorylation sites can become transiently exposed during co-translational protein folding, supporting their potential for modification in intermediate conformational states. These simulations suggest that cryptic modifications can influence protein folding, expression, and stability. Experimental validation using western blotting and protein half-life measurements confirmed that phosphomimetic substitutions at cryptic sites reduce protein expression, reinforcing their hypothesis that such modifications affect protein stability. Furthermore, overlap between their annotated cryptic sites and cancer-associated mutations from multiple databases highlights their functional relevance, particularly in contexts such as tumor fitness and disease progression. From evolutionary, computational, structural, and experimental perspectives, this work establishes a comprehensive framework for studying cryptic PTMs and opens avenues to explore how activation of cryptic signaling pathways may contribute to disease mechanisms.

The authors have addressed the comments and suggestions satisfactorily. The clarification regarding the statistical testing of entropy score distributions, the additional analyses comparing cryptic and non-cryptic sites in the context of mutation overlap and functional relevance, responses related to the SPECTRUS methodology and the use of pLDDT/PAE scores are reasonable. Overall, the revisions improve the clarity of the study.

Referee #2:

This manuscript uses bioinformatic, computational biophysical, and experimental biochemical approaches to explore the significance of so-called "cryptic phosphosites" found within the hydrophobic cores of proteins. By correlating predicted structure with the presence of phosphorylated residues documented in phosphosite databases, the authors compile a list of thousands of cryptic sites that are not surface accessible in the native folded state. They go on to show for selected examples that such sites are potentially accessible in transient folding intermediates, and that phosphomimetic mutation of these sites leads to decreased protein expression. They hypothesize that such sites could play a critical role in regulating normal proteostasis, which would be a novel and important finding. The identification of several such cryptic sites in cancer-associated mutations in tumor suppressors provides additional interest.

Overall, I believe this careful and thorough analysis is an important advance in our understanding of mechanisms that regulate protein abundance, and may identify a novel point of regulation. While I don't necessarily agree with all the authors' interpretations of their results, I do think the work is worth sharing widely so that others can build on their results.

Many of my comments and concerns with the original preprint version of the manuscript have been addressed either in the authors' detailed rebuttal or in the revised manuscript. However, I do still have a few suggestions that would improve the manuscript if addressed.

1. In the introduction, the authors write that, "In this manuscript, we combine experiments, theory and simulations to tackle the central question of whether specific non-native, folding-associated protein conformers could play physiological functions in the protein regulation machinery." I believe this is misleading, almost as if it was borrowed from another manuscript. The focus here is clearly on the role of phosphorylation of sites predicted to be buried in the native structure.
2. I strongly believe that the dynamical filtering used to remove sites that might be momentarily exposed during thermal fluctuations and interactions with other molecules is a mistake, as those sites are precisely the ones that would have the highest likelihood of being involved in regulation of protein abundance and/or activity as a result of phosphorylation. One could argue that these would be the most interesting and biologically relevant sites. By filtering them out, the authors essentially limit their analysis to co-translational phosphorylation. I particularly find the use of the term "false positive" to be unfortunate, as it implies that such sites are not real or potentially important. While I would not require that the authors do additional analyses that include these potentially dynamically accessible sites, I would like to see some discussion of the issue and an acknowledgment that they are limiting their analysis to sites that must be phosphorylated cotranslationally.
3. The analysis of protein abundance/stability of native and phosphomimetic mutants of cryptic sites (Fig. 5) uses the term "half-life" inappropriately in my view. Their model is that co-translational phosphorylation targets the protein for almost immediate degradation, which makes sense. But if that is true, essentially all of the phosphomimetic mutant protein will be targeted for degradation as soon as it is synthesized, so the "half-life" is actually the half-time for a partially folded protein to be degraded by the proteolytic machinery. Importantly, no functional protein should ever be present for these mutants. By contrast, the timecourse for the wt protein should be the sum of the curves for the alanine mutant and the phosphomimetic mutants. I would recommend the text be modified to address this.
4. The discussion of the evolutionary conservation of potential cryptic phosphosites should explicitly acknowledge the very different evolutionary history of phosphotyrosine vs. phosphoserine and phosphothreonine (pTyr signaling first appearing around the time of the emergence of the metazoan lineage).
5. It is unsurprising that phosphomimetic mutations at cryptic sites of tumor suppressors are found in cancer, given the expected destabilization of the native folded state for such mutations. Can the authors point to any specific examples of increased phosphorylation of such sites leading to decreased expression in cancer? Are phosphomimetic mutations at cryptic sites found more frequently than expected compared with other mutations expected to destabilize the native folded structure?

Referee #3:

Gasparotto et al provided a revised manuscript, with several useful "reviewer only" explanations in the rebuttal. Since the reviewer comments are published along with the paper in EMBOJ, this info (tabs and figures) will be available. In this case this is particularly useful as the manuscript revisions are more on the moderate side. What let the authors changes figure 4 into pie charts is not clear to me. With regards to 2.8., it is of course possible to shuffle P-sites on proteins in a distribution-controlled manner and check whether main findings hold.

Furthermore reviewer 1 and myself argue that separation of tyrosine and serine/threonine is important, as they are fundamentally different with respect to distribution in domains (95% of P-tyrosines are in domains) vs disordered regions (95% of P-S/T are in disordered regions) and with respect to solvent exposure. However, the authors do not find this difference (actually any differences between P-S/T and P-Y) worth mentioning in the main text.

In summary, I think it is an important, original analysis of buried PTM sites with interesting insights and the revisions started from a submission I anyways liked a lot in the first place. I want to congratulate the authors to their work.

Rev_Com_number: RC-2025-02892

New_manu_number: EMBOJ-2025-121567-T

Corr_author: Biasini

Title: Mapping Cryptic Phosphorylation Sites in the Human Proteome

**UNIVERSITÀ
DI TRENTO**

**Dipartimento di
Biologia Cellulare, Computazionale e Integrata**

EMILIANO BIASINI, Ph.D.
Associate Professor
Dulbecco Telethon Laboratory of Prions and Amyloids
Dipartimento CIBIO, Università di Trento
Room Red#2, Via Sommarive 9, 38123 Trento (ITALY)
Email: emiliano.biasini@unitn.it
Ufficio: +39-0461-282-740; Lab: +39-0461-283-665

Trento, July 25, 2025

TO:
Hartmut Vodermaier, PhD
Senior Editor, The EMBO Journal

Dear Dr. Vodermaier,

Thank you very much for your positive assessment and for treating our revised manuscript as a regular EMBO Journal revision. We are grateful to you and the referees for the thoughtful feedback and are pleased that the reviewers are satisfied with our revisions. We have now carefully addressed the remaining presentational changes and suggestions from the reviewers to finalize the manuscript for publication.

Sincerely,

Prof. Emiliano Biasini
(on behalf of all co-authors)

Point-by-point rebuttal

Referee #1:

This study categorizes cryptic and non-cryptic post-translational modifications (PTMs) and investigates their roles in regulating protein activity and signaling pathways. By integrating AlphaFold-predicted structures, solvent accessibility data, and stringent criteria to reduce false positives, the authors systematically identified and validated cryptic PTM sites. Through molecular dynamics (MD) simulations, they demonstrated that cryptic phosphorylation sites can become transiently exposed during co-translational protein folding, supporting their potential for modification in intermediate conformational states. These simulations suggest that cryptic modifications can influence protein folding, expression, and stability. Experimental validation using western blotting and protein half-life measurements confirmed that phosphomimetic substitutions at cryptic sites reduce protein expression, reinforcing their hypothesis that such modifications affect protein stability. Furthermore, overlap between their annotated cryptic sites and cancer-associated mutations from multiple databases highlights their functional relevance, particularly in contexts such as tumor fitness and disease progression. From evolutionary, computational, structural, and experimental perspectives, this work establishes a comprehensive framework for studying cryptic PTMs and opens avenues to explore how activation of cryptic signaling pathways may contribute to disease mechanisms. The authors have addressed the comments and suggestions satisfactorily. The clarification regarding the statistical testing of entropy score distributions, the additional analyses comparing cryptic and non-cryptic sites in the context of mutation overlap and functional relevance, responses related to the SPECTRUS methodology and the use of pLDDT/PAE scores are reasonable. Overall, the revisions improve the clarity of the study.

We sincerely thank Referee#1 for the thoughtful and thorough evaluation of our work, and for acknowledging the significance of our revisions in enhancing the clarity and impact of the study.

Referee #2:

This manuscript uses bioinformatic, computational biophysical, and experimental biochemical approaches to explore the significance of so-called "cryptic phosphosites" found within the hydrophobic cores of proteins. By correlating predicted structure with the presence of phosphorylated residues documented in phosphosite databases, the authors compile a list of thousands of cryptic sites that are not surface accessible in the native folded state. They go on to show for selected examples that such sites are potentially accessible in transient folding intermediates, and that phosphomimetic mutation of these sites leads to decreased protein expression. They hypothesize that such sites could play a critical role in regulating normal proteostasis, which would be a novel and

important finding. The identification of several such cryptic sites in cancer-associated mutations in tumor suppressors provides additional interest.

Overall, I believe this careful and thorough analysis is an important advance in our understanding of mechanisms that regulate protein abundance, and may identify a novel point of regulation. While I don't necessarily agree with all the authors' interpretations of their results, I do think the work is worth sharing widely so that others can build on their results.

Many of my comments and concerns with the original preprint version of the manuscript have been addressed either in the authors' detailed rebuttal or in the revised manuscript. However, I do still have a few suggestions that would improve the manuscript if addressed.

1. In the introduction, the authors write that, "In this manuscript, we combine experiments, theory and simulations to tackle the central question of whether specific non-native, folding-associated protein conformers could play physiological functions in the protein regulation machinery." I believe this is misleading, almost as if it was borrowed from another manuscript. The focus here is clearly on the role of phosphorylation of sites predicted to be buried in the native structure.

We thank Referee#2 for pointing this out. We have modified the sentence in the introduction to reflect the focus of the study more accurately. It now reads:

"In this manuscript, we combine experiments, theory, and simulations to investigate whether phosphorylation at sites predicted to be buried in the native protein structure can occur co-translationally and influence protein expression and stability." (lane 105)

2. I strongly believe that the dynamical filtering used to remove sites that might be momentarily exposed during thermal fluctuations and interactions with other molecules is a mistake, as those sites are precisely the ones that would have the highest likelihood of being involved in regulation of protein abundance and/or activity as a result of phosphorylation. One could argue that these would be the most interesting and biologically relevant sites. By filtering them out, the authors essentially limit their analysis to co-translational phosphorylation. I particularly find the use of the term "false positive" to be unfortunate, as it implies that such sites are not real or potentially important. While I would not require that the authors do additional analyses that include these potentially dynamically accessible sites, I would like to see some discussion of the issue and an acknowledgment that they are limiting their analysis to sites that must be phosphorylated cotranslationally.

We agree with Referee#2 that dynamically accessible sites, such as those transiently exposed during thermal fluctuations or through interactions with other molecules, may play significant roles in post-translational regulation, and we appreciate the opportunity to clarify our rationale and limitations. Our filtering strategy was intentionally designed to focus on phosphorylation events that could occur during protein synthesis, i.e., co-translationally, when the ribosome still constrains the nascent chain. In this context, we

aimed to minimize the inclusion of sites with exposure likely to result from post-translational dynamics rather than folding-associated intermediates. We recognize that using the term "false positive" may have inadvertently implied a lack of biological relevance, which was not our intention. We have revised the text to use more neutral language (e.g., "post-translationally exposed sites") and now explicitly acknowledge in the discussion that this filtering narrows our analysis to a specific subset of potential regulatory sites.

3. The analysis of protein abundance/stability of native and phosphomimetic mutants of cryptic sites (Fig. 5) uses the term "half-life" inappropriately in my view. Their model is that co-translational phosphorylation targets the protein for almost immediate degradation, which makes sense. But if that is true, essentially all of the phosphomimetic mutant protein will be targeted for degradation as soon as it is synthesized, so the "half-life" is actually the half-time for a partially folded protein to be degraded by the proteolytic machinery. Importantly, no functional protein should ever be present for these mutants. By contrast, the timecourse for the wt protein should be the sum of the curves for the alanine mutant and the phosphomimetic mutants. I would recommend the text be modified to address this.

We agree that the term "half-life," traditionally used to describe the decay of fully folded, functional proteins, may not accurately capture the dynamics in our system, particularly for the phosphomimetic mutants. As the reviewer correctly points out, our model suggests these mutants are targeted for degradation co-translationally or shortly after synthesis, before reaching their native, functional conformation. We have therefore revised the text to clarify that, in this context, the measured decay reflects the clearance kinetics of newly synthesized, potentially misfolded or incompletely folded polypeptides, rather than the half-life of mature protein species. We now describe this parameter throughout the text as "half-time" or "clearance rate".

4. The discussion of the evolutionary conservation of potential cryptic phosphosites should explicitly acknowledge the very different evolutionary history of phosphotyrosine vs. phosphoserine and phosphothreonine (pTyr signaling first appearing around the time of the emergence of the metazoan lineage).

We have revised the discussion to explicitly acknowledge the distinct evolutionary trajectories of phosphotyrosine (pTyr) compared to phosphoserine and phosphothreonine (pSer/pThr). Specifically, we now note that while pSer and pThr signaling are ancient and broadly conserved across eukaryotes, pTyr signaling is thought to have emerged later, around the origin of the metazoan lineage, in association with the expansion of tyrosine kinases and SH2-domain-containing proteins. This divergence has important implications for interpreting the evolutionary conservation of cryptic phosphosites, particularly tyrosine residues, and may explain differences in selective pressures and functional relevance across lineages (lane 361). We thank the reviewer for highlighting this important point, improving our conservation data interpretation.

5. It is unsurprising that phosphomimetic mutations at cryptic sites of tumor suppressors are found in cancer, given the expected destabilization of the native folded state for such mutations. Can the authors point to any specific examples of increased phosphorylation of such sites leading to decreased expression in cancer? Are phosphomimetic mutations at cryptic sites found more frequently than expected compared with other mutations expected to destabilize the native folded structure?

We appreciate comments. We agree that phosphomimetic mutations at cryptic sites in tumor suppressors are consistent with the expectation that such substitutions may destabilize protein folding and reduce steady-state expression, potentially contributing to loss-of-function phenotypes in cancer. In response to the specific question, we have now included in the revised discussion a few specific examples from the literature where increased phosphorylation at predicted cryptic sites has been associated with decreased protein stability or expression in cancer-relevant contexts. One notable case that we now mention in the text is the phosphorylation of cryptic serine residues in the tumor suppressor PTEN, which has been reported to reduce its stability and membrane localization (e.g., Vazquez et al., *Mol Cell*, 2000) (lane 315). Regarding the second question, while our current work highlights the functional impact of phosphomimetic substitutions at cryptic sites, a systematic comparison with other destabilizing mutations, such as hydrophobic-to-polar substitutions or proline insertions, has not yet been performed. Addressing this will involve a more extensive cross-referencing of cryptic phosphosites with cancer mutation databases enriched with phenotypic annotations, such as tumor type, progression, and protein expression data. This effort will be the subject of a forthcoming study to determine whether phosphomimetic mutations at cryptic sites are found more frequently than expected and whether they exhibit distinct functional or clinical associations compared to other destabilizing alterations.

Referee #3:

Gasparotto et al provided a revised manuscript, with several useful "reviewer only" explanations in the rebuttal. Since the reviewer comments are published along with the paper in EMBOJ, this info (tabs and figures) will be available. In this case this is particularly useful as the manuscript revisions are more on the moderate side. What let the authors changes figure 4 into pie charts is not clear to me. With regards to 2.8., it is of course possible to shuffle P-sites on proteins in a distribution-controlled manner and check whether main findings hold. Furthermore reviewer 1 and myself argue that separation of tyrosine and serine/threonine is important, as they are fundamentally different with respect to distribution in domains (95% of P-tyrosines are in domains) vs disordered regions (95% of P-S/T are in disordered regions) and with respect to solvent exposure. However, the authors do not find this difference (actually any differences between P-S/T and P-Y) worth mentioning in the main text. In summary, I think it is an important, original analysis of buried PTM sites with interesting insights and the revisions started from a submission I anyways liked a lot in the first place. I want to congratulate the authors to their work.

UNIVERSITÀ
DI TRENTO

Dipartimento di
Biologia Cellulare, Computazionale e Integrata

We thank Referee#3 for the positive and constructive feedback and support throughout the review process. In Figure 4, we replaced the pie charts with dot plots to provide a more intuitive and immediately interpretable visualization of the proportional distribution of cryptic versus non-cryptic PTMs across different protein classes. We changed the figure legend accordingly (lane 966). As for point 2.8, shuffling P-sites in a distribution-controlled manner would provide an informative complementary control; we will consider this suggestion a potential follow-up analysis that could help assess the robustness of the observed enrichments. Finally, we have extended the description of the distributions to explicitly discuss the differential characteristics of phosphotyrosine versus phosphoserine/threonine sites, particularly about their domain localization and solvent accessibility (lane 157).

Comment from data editors:

1) Please note that the box plots need to be defined in terms of minima, maxima, centre, bounds of box and whiskers, and percentile in the legends of figures 4A-C

We have inserted all the indicated values into the legend of Figure 4A-C

2) Please note that information related to N is missing in the legends of figures 4A-C

We have also added the values for N into the legend of Figure 4A-C